# MODEL PREDICTIVE ADVERSARIAL IMITATION LEARNING FOR PLANNING FROM OBSERVATION

**Tyler Han, Yanda Bao, Bhaumik Mehta, Gabe Guo, Sanghun Jung,**
**Anubhav Vishwakarma, Emily Kang, Rosario Scalise, Jason Zhou,**
**Bryan Xu, Byron Boots**
Paul G. Allen School of Computer Science & Engineering
University of Washington

## ABSTRACT

Humans can often perform a new task after observing a few demonstrations by inferring the underlying intent. For robots, recovering the intent of the demonstrator through a learned reward function can enable more efficient, interpretable, and robust imitation through planning. A common paradigm for learning how to plan-from-demonstration involves first solving for a reward via Inverse Reinforcement Learning (IRL) and then deploying it via Model Predictive Control (MPC). In this work, we unify these two procedures by introducing planning-based Adversarial Imitation Learning, which simultaneously learns a reward and improves a planning-based agent through experience while using observation-only demonstrations. We study advantages of planning-based AIL in generalization, interpretability, robustness, and sample efficiency through experiments in simulated control tasks and real-world navigation from few or single observation-only demonstration.

## 1 INTRODUCTION

Inverse Reinforcement Learning (IRL) offers a principled approach to imitation learning by inferring the underlying intent, or reward function, that explains expert behavior. A fundamental advantage of IRL is that this reward is often readily generalizable beyond the support of the demonstration data, enabling the discovery of new policies through interaction and plans through self-prediction. Especially when demonstrations are sparse, ambiguous, or suboptimal, IRL's interpretability is particularly well-suited for domains where understanding preferences and ensuring reliable planning are essential, such as routing on Google Maps (Barnes et al., 2023), socially aware navigation (Kretzschmar et al., 2016), and autonomous driving (Bronstein et al., 2022).

For real-time systems, learned IRL and Inverse Optimal Control (IOC) rewards are typically deployed via Model Predictive Control

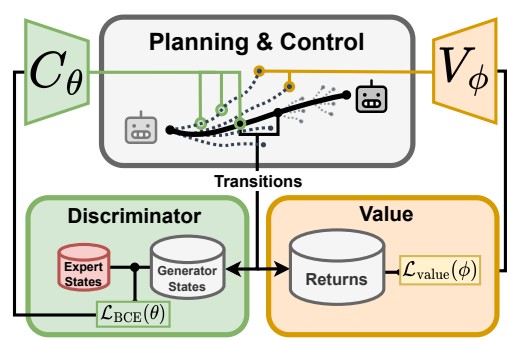

Figure 1: Model Predictive Adversarial Imitation Learning (MPAIL) learns costs for a planning-based, Model Predictive Control (MPC) agent from observation-only demonstration. Interactions with these costs are simultaneously used to learn a value function for experience-based reasoning beyond the horizon of the planner.

(MPC) (Lee et al., 2022b; Rosbach et al., 2019; Triest et al., 2023; Das et al., 2021; Lee et al., 2021; Kuderer et al., 2015; Lee et al., 2022a). Here, the offline IRL algorithm iteratively solves a Reinforcement Learning (RL) problem in an inner loop, guided by the current reward estimate. An outer loop then updates this reward to minimize the discrepancy between the agent's and the expert's behavior. Once training is complete, the resulting reward is integrated with MPC for real-time planning and control.

Adversarial Imitation Learning (AIL) has made significant improvements over IRL in algorithmic complexity and sample efficiency (Ho & Ermon, 2016; Baram et al., 2017). However, the reliance on an RL policy in AIL methods complicates their use in applications with safety constraints (Lee et al., 2022b; Triest et al., 2023; Das et al., 2021). Further limited by partial observability, these deployments will often prioritize planning using a model for the sake of real-time performance, trustworthiness, and interpretability (Han et al., 2024a; Katrakazas et al., 2015; Choudhury et al., 2018).

In this work, we derive *planning-based* AIL, yielding key benefits:

1. **Planning-from-Observation (PfO).** Towards interpretable yet scalable imitation learning, a predictive model precludes the need for expert action data and enables access to the agent's optimization landscape. This grants crucial insight and steerability into the agent's decision making process even as it learns from ambiguous expert data. *We further show that this improves on out-of-distribution generalization, robustness, and sample efficiency when compared to policy-based AIL. We also demonstrate how policy-based AIL is fundamentally limited by the absence of reward deployment.*

2. **Unification of IRL and MPC.** Otherwise considered independent training and deployment procedures, planning-based AIL allows for end-to-end interactive learning of the entire planner. Critical online settings (e.g., dynamics, preferences, control constraints) can thus be brought into training while enabling experience-based reasoning beyond the planning horizon, which we demonstrate in this work. *We also find this induces a more effective adversarial dynamic than policy-based generators when learning from partial observations in the real world.*

To our knowledge, this work presents the first end-to-end planning-from-observation (PfO) framework, extending PfO to continuous spaces and interactive learning. By choosing Model Predictive Path Integral control (MPPI) (Williams et al., 2017) as the embedded planner, we further gain theoretical perspective on planning-based AIL and its relationship to the seminal GAIL objective (Ho & Ermon, 2016; Torabi et al., 2019a). Thus, we name this learning algorithm: Model Predictive Adversarial Imitation Learning (MPAIL {*impale*}).

## 2 RELATED WORK

**IRL-MPC.** High-dimensional continuous control applications often require an online planner for real-time control, trustworthiness, safety, or additional constraints. When using IRL to learn a reward, online deployments of these reward functions rely on an independent online MPC procedure. To enable learning local costmaps across perception and control for off-road navigation, Lee et al. (2022b) and Triest et al. (2023) similarly propose solving the forward RL problem using MPPI but deploy the reward on a different configuration more suitable for real-time planning and control. This reward deployment framework of *IRL-then-MPC* is currently the dominant approach for planning in high-dimensional continuous control tasks from demonstration (Lee et al., 2022b; Rosbach et al., 2019; Triest et al., 2023; Das et al., 2021; Lee et al., 2021; Kuderer et al., 2015; Lee et al., 2022a).

**Model-Based IRL and Planning-Based RL.** Various other works have also explored model-based AIL (Baram et al., 2016; Bronstein et al., 2022; Sun et al., 2021). However, scope is directed at training stabilization and policy optimization rather than examining planning with the learned reward. When the reward is known, as in RL, planning-based algorithms have demonstrated considerable improvement in simulation benchmarks over existing state-of-the-art RL algorithms through developments such as: online trajectory optimization, value bootstrapping, latent state planning, policy-like or learned sampling priors, and much more (Hansen et al., 2024; Bhardwaj et al., 2021; Lowrey et al., 2019; Jawale et al., 2024). This work's implementation of MPAIL performs online trajectory optimization and value bootstrapping. This work *does not* implement latent state planning nor a policy-based prior to better isolate our investigations in interpretability and planning (Fu et al., 2018; Sun et al., 2021). Other existing integrations of planning-based RL with imitation learning also rely on access to expert actions (Li et al., 2025; Yin et al., 2022). No aforementioned works, save (Jawale et al., 2024), evaluate planning-based RL or AIL on a real world platform. However, we find that it is precisely real-world and out-of-distribution settings in which planning and control is most advantageous.

## 3 Model Predictive Adversarial Imitation Learning

### 3.1 The POMDP Setting and the Model Predictive Agent

We adopt the Partially Observable Markov Decision Process (POMDP) to best consider highly desirable applications of IRL in which partial observability and model-based planning play crucial roles. In an unknown world state $s_w \in \mathcal{S}_w$, the agent makes an observation $o \sim p(o|s_w)$. From a history of observations $\mathbf{o}_{0:t}$, the Agent perceives its *state* $s_t \sim p(s|\mathbf{o}_{0:t})$. *Actions* $a \in \mathcal{A}$ and states $s \in \mathcal{S}$ together allow the agent to self-predict forward in time using its *predictive model* $f : \mathcal{S} \times \mathcal{A} \rightarrow \mathcal{S}$. Note that these definitions crucially suggest the partial observability of $s$ due to the implicit dependency on the observation history $\mathbf{o}_{0:t}$ through the agent's perception (e.g. mapping (Jung et al., 2024)). However, partial observations in $\mathcal{S}$ are desirably used for demonstrations to perform IRL and AIL, as full observation history would quickly become intractable (Triest et al., 2023).

The planner itself is a model predictive agent. It is capable of performing *model rollouts* $\tau_t^{(H)} = \{s_{t'}, a_{t'}\}_{t'=t}^{t+H}$ such that $s_{t'+1} = f(s_{t'}, a_{t'})$, and each rollout can thus be evaluated with a cost $C(\tau_t)$. The agent's objective is to create an $H$-step action sequence $\mathbf{a}_{t:t+H}$, or *plan*, that best minimizes the plan's corresponding trajectory cost. The true costs or rewards under which the expert is acting is not known. Towards learning-from-observation (LfO) and lower-level policies, we consider expert data in which only states are available, as actions may be challenging or impossible to observe (Torabi et al., 2019b).

### 3.2 Adversarial Imitation Learning from Observation

IRL algorithms aim to learn a cost function that minimizes the cost of expert trajectories while maximizing the cost of trajectories induced by other policies (Torabi et al., 2019a; Ho & Ermon, 2016). As the problem is ill-posed and many costs can correspond to a given set of demonstrations, the principle of maximum entropy is imposed to obtain a uniquely optimal cost. It can be shown that the entropy maximizing distribution is a Boltzmann distribution (Ziebart et al.). The state-only IRL from observation problem can be formulated by costing state-transitions $c(s, s')$ rather than state-actions $c(s, a)$ as in (Torabi et al., 2019a):

$$\text{IRLfO}_\psi(\pi_E) = \arg\max_{c \in \mathbb{R}^{\mathcal{S} \times \mathcal{S}}} -\psi(c) + \left( \min_{\pi \in \Pi} -\lambda \mathbb{H}(\pi) + \mathbb{E}_\pi[c(s, s')] \right) - \mathbb{E}_{\pi_E}[c(s, s')], \qquad (1)$$

where $\psi(c)$ is a convex cost regularizer, $\pi_E$ is the expert policy, $\mathbb{H}(\cdot)$ is the entropy, and $\Pi$ is a family of policies.

As shown in (Ho & Ermon, 2016; Torabi et al., 2019a), this objective can be shown to be dual to the Adversarial Imitation Learning (AIL) objective under a specific choice of cost regularizer $\psi$,

$$\min_{\pi \in \Pi} \max_{D \in [0,1]^{\mathcal{S} \times \mathcal{S}}} \mathbb{E}_\pi[\log(D(s, s'))] + \mathbb{E}_{\pi_E}[\log(1 - D(s, s'))] - \lambda \mathbb{H}(\pi), \qquad (2)$$

where $D(\cdot)$ is the discriminator function. The exact form of $D$ has consequences on the policy objective and differs by AIL algorithm. Now equipped with the optimization objective, we continue in our derivation of planning-based AIL by choosing the form of the policy class and reward function.

### 3.3 Choosing a Policy and Reward

To reiterate, we set our sights on the AIL objective (Equation (2)) which aims to simultaneously learn reward and policy from demonstration. However, the formulation remains intimately connected with policy optimization through the assumption of an RL procedure. Towards planning-based optimization, we proceed by modifying the RL formulation in Equation (1) as described in (Ho & Ermon, 2016; Fu et al., 2018). Similar to (Bhardwaj et al.), we replace the entropy loss $-\lambda \mathbb{H}(\pi)$ with a Kullback-Leibler (KL) divergence constraint on the previous policy $\bar{\pi}$:

$$\min_{\pi \in \Pi} \mathbb{E}_\pi[c(s, s')] + \beta \, \mathbb{KL}(\pi \,||\, \bar{\pi}). \qquad (3)$$

Note that this incorporates prior information about the policy (i.e. previous plans) while seeking the next maximum entropy policy. Specifically, as shown in Section B.1, the closed form solution to

---

**Algorithm 1** Model Predictive Adversarial Imitation Learning

---

**Require:** Expert state-transitions $\mathcal{D}_E = \{(s, s')\}$
    Maximum-Entropy Planner $\pi_{\text{MPPI}}$, Discriminator $D_\theta$, Value $V_\phi$
1: **while** not converged **do**
2:    Collect transitions $(s, s', r_\theta(\cdot)) \in d^\pi$ by running $\pi_{\text{MPPI}}$ (Alg. 3) in the environment
3:    Update Discriminator parameters $\theta$ with Binary Cross Entropy (BCE) loss:

$$\nabla_\theta \mathbb{E}_{s,s' \sim d^\pi}[\log(D_\theta(s, s'))] + \nabla_\theta \mathbb{E}_{s,s' \sim d^{\pi_E}}[\log(1 - D_\theta(s, s'))] \tag{5}$$

4:    Update Value parameters $\phi$ using estimated returns:

$$\nabla_\phi \mathbb{E}_{s \sim d^\pi}[(G_t - V_\phi(s))^2] \tag{6}$$

5: **end while**

---

Equation (3) is $\pi^*(a|s) \propto \overline{\pi}(a|s)e^{\frac{-1}{\beta}\overline{c}(s,a)}$, where $\overline{c}(s, a) = \sum_{s' \in \mathcal{S}} \mathcal{T}(S_{t+1} = s'|S_t = s)c(s, s')$ and $\mathcal{T}(S_{t+1} = s'|S_t = s)$ denotes the transition probability from state $s$ to $s'$.

We then observe that a choice of planner satisfies the RL objective as in Equation (3). By choosing Model Predictive Path Integral (MPPI) as the planner, as proven in Section B.1, we solve an equivalent problem provided the MDP is uniformly ergodic. Namely, MPPI solves for a KL-constrained cost-minimizer over trajectories (Bhardwaj et al.):

$$\min_{\pi \in \Pi} \mathbb{E}_{\tau \sim \pi}[C(\tau) + \beta \, \mathbb{KL}(\pi(\tau) \,||\, \overline{\pi}(\tau))] \tag{4}$$

where $C(\tau)$ is the discounted cost of a trajectory and $\mathbb{KL}(\pi(\tau)||\overline{\pi}(\tau))$ is the discounted KL divergence over a trajectory.

Model rollouts must practically be limited to some timestep length $H$, resulting in myopic plans and limiting applications to short-horizon tasks (Bhardwaj et al., 2021). To resolve this, infinite-horizon MPPI bootstraps the final states in the rollout using a terminal cost-to-go, or value, function (Bhardwaj et al., 2021; Hatch & Boots, 2021; Lowrey et al., 2019) $V_\phi : \mathcal{S} \to \mathbb{R}$ that estimates the expected return $G_t$ of a state $s_t$ as $G_t = \mathbb{E}_\pi[R_{t+1} + \gamma R_{t+2} + ...|S_t = s_t]$, where $R_{t+1} = R(s_t, s_{t+1})$. The result of MPPI's approximately global policy optimization at each timestep is what is referred to as the *MPPI policy*, $\pi_{\text{MPPI}}$. Section B.1 proves how this formulation can be equivalent to the entropy-regularized RL objective, while also in the observation-only setting. Pseudocode for the MPPI procedure can be found in Algorithm 2 as well as for its adaptation as an RL policy in Algorithm 3. Figure 2 illustrates the policy.

Our chosen AIL agent, infinite-horizon MPPI, can be viewed as optimizing for a new policy at every state. As in the online learning perspective (Wagener et al., 2019), these MPPI optimizations can occur online rather than offline as part of, for instance, a slower actor-critic update. By deconstructing the agent this way, we require not the policy to generalize but the reward.

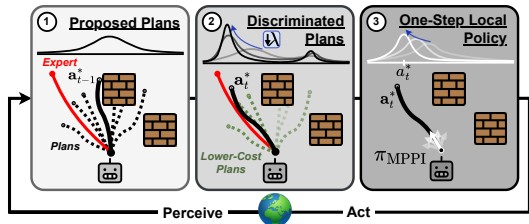

Figure 2: Illustration of $\pi_{\text{MPPI}}$ in MPAIL. (1) A set of action sequences (plans) are sampled and rolled out. (2) Plans are costed according to the discriminator, shifting the distribution towards the expert. Temperature $\lambda$ optionally decays over episodes, narrowing the distribution. (3) The policy $\pi_{\text{MPPI}}$ is the result of a Gaussian fit to the optimized plans and their respective first actions.

Provided the agent, we now proceed with selecting its objective. Recent work has shown many potential choices of valid policy objectives, each with various empirical trade-offs (Orsini et al., 2021). We found the reward as defined in Adversarial Inverse Reinforcement Learning (AIRL) (Fu et al., 2018) to be most stable when combined with the value function when applied to infinite-horizon MPPI. In the state-only setting, the policy objective becomes $r(s, s') = \log(D(s, s')) - \log(1 - D(s, s'))$ and the discriminator $D(s, s') = \sigma \circ d_\theta(s, s')$. Simply put, the reward is the logit of the discriminator $r(s, s') = d_\theta(s, s')$.

In short, our choice of $\pi_{\text{MPPI}}$ and $r(s, s')$ yields MPAIL. As proven in Section B.2, MPAIL is indeed an AIL algorithm in the sense that it minimizes divergence from the expert policy. The procedure (Algorithm 1) itself closely resembles the original GAIL procedure. However, upon updating the value network, MPAIL does not require a policy update thereafter, since "policies" are in theory solved online. In practice a temperature decay can be helpful for preventing early collapse, especially in the case of online model learning. We leave a theoretical justification for this choice for future work. An overview of the training procedure can be found in Figure 1. A discussion of further meaningful implementation details, like spectral normalization, can be found in Section C. Though, these modifications are kept to a minimum towards our analysis of $\pi_{\text{MPPI}}$ in AIL.

## 4 EXPERIMENTAL RESULTS

Advancements in IRL and AIL continue to demonstrate improvements in sample efficiency. However, there remains an apparent gap with applications to robot learning. In addition to fundamental investigations about the MPAIL algorithm itself, our experiments target evaluations critical towards real-world robustness and generalization.

Without a policy, vanilla MPPI possesses no "memory" about actions taken in previous episodes, save for those implied through the value $V_\phi(s)$. And as discussed in Section 3, MPPI's approximate online optimization is more practical for robustness and generalization but potentially less accurate for producing policies. These novelties raise a critical question about planning-based AIL; are policies solved online through planning sufficient as adversarially generative policies? We find that MPAIL indeed trains an effective imitator, provoking our follow-up questions:

**Q1** What is the advantage of deploying an AIL planner over an AIL policy?

**Q2** How does MPAIL help enable real-world planning capabilities from observation?

**Q3** How does MPAIL compare to existing AIL algorithms?

Hyperparameter settings are kept consistent across all experiments. Exact values and other implementation details such as regularization and computation are reported and discussed in Section C.

SIMULATED NAVIGATION TASK

For simulated evaluation, we design a navigation task with a 10-DoF vehicle. Reward is proportional to the negative squared distance to $(10, 10)$. Initial poses are within 1 m of $(0, 0)$. *The state is 12-dimensions: position, orientation, linear velocity, and angular velocity*. Actions include target velocity and steering angle. MPAIL plans using an approximate prior model, the Kinematic Bicycle Model (Han et al., 2024b). This approximate model is not tuned to the agent dynamics. For instance, slipping and suspension dynamics occur in simulation but are unmodeled.

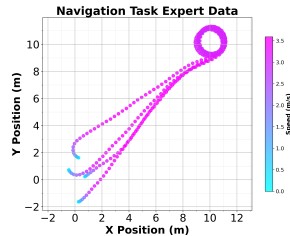

Figure 3: **Four Expert Trajectories in Navigation Task.** Cars initialized around $(0, 0)$.

The expert for this environment is obtained through running PPO on a known reward (see Section D.2 for details) (Schulman et al., 2017b). At convergence, the optimal policy occasionally circles near the goal instead of remaining on it (see Figure 3). We choose to use the circling demonstrations as expert data, because it is a more challenging behavior to imitate. This is corroborated by (Orsini et al., 2021), who stress that demonstrator suboptimality and multimodality is a critical component in algorithm evaluation towards practical AIL from human data.

This circling behavior acts as a critical "distractor mode". In training, the policy may begin to only circle around the origin. If the AIL algorithm is not able to sufficiently explore, training collapses such that the policy continuously circles the origin, unable to return to the expert distribution which requires the circling behavior to occur around the goal. For instance, we find that AIRL is unable to successfully learn both behaviors likely due to the instability introduced by its logit shift (Orsini et al., 2021).

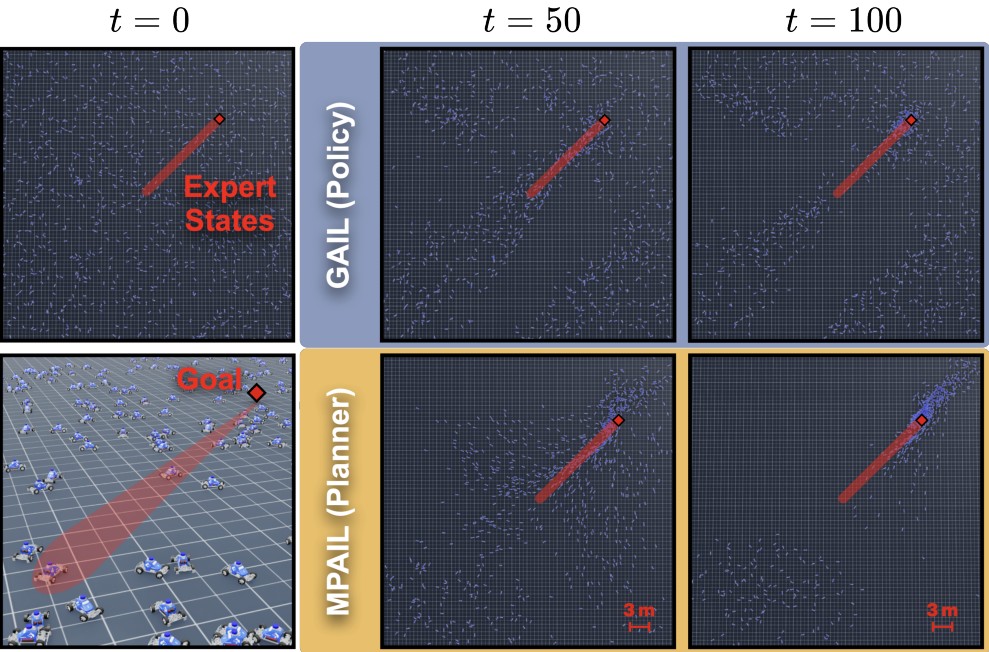

Figure 4: **Comparison of policy-based and planning-based AIL in Out-of-Distribution (OOD) states.** Agents trained on the navigation task (Section 4) are placed uniformly with random orientation between a $40 \times 40$ m box centered on $(0,0)$. The policy and planner are run for 100 timesteps in the environment. Data support of the expert exists mainly between $(0,0)$ and $(10,10)$[1]. Quantitative evaluation of this experiment can be found in Figure 5. A comparison which includes a learned dynamics model can be found in Figure 8

.

## 4.1 OUT-OF-DISTRIBUTION ROBUSTNESS THROUGH PLANNING – Q1

When deploying learning-based methods to the real-world, reliable performance in out-of-distribution (OOD) states are of critical importance, especially in imitation learning when expert data can be extremely sparse. We show that planning-based AIL (MPAIL) improves generalization capabilities when OOD. In this experiment, we use the simulated navigation environment but *expand the region of uniformly distributed initial positions and orientations from a 1×1 square to a large 40×40 m square around the origin* (Figure 4). The policy-based approach is represented by GAIL, as AIRL does not meaningfully converge in the navigation task (Section 4). Only four expert trajectories are used in training.

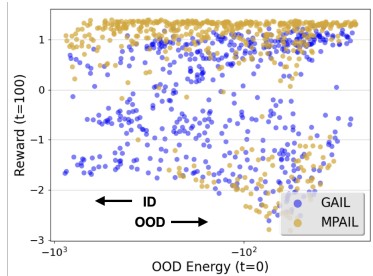

Figure 5: **OOD Navigation Evaluation.** Agent initial poses vary from In-distribution (ID) to OOD relative to the expert data and are plotted with their final reward after 100 timesteps. Metric from (Liu et al., 2020) (see Section D.2).

We find that planning-based AIL generalizes to significantly more states than policy-based AIL when outside the support of expert data. In this experiment, the planner's horizon is a maximum of 3 meters. As a result, the task horizon may be up to 15 times longer than the planning horizon. Evidently, a planner could not navigate to the goal if the learned optimization landscape (induced by cost $c_\theta$ and value $V_\phi$) did not also generalize to OOD states. These results suggest a fundamental limitation of current AIL approaches and their single policy solution. The trained reward and value are inefficiently underutilized in policy-based AIL and not utilized at all on deployment. By contrast, *MPAIL re-introduces the reward and value online to solve for new policies*

---

[1]Note that the state space is 12-dimensional; expert data support is extremely sparse in this environment.

*each moment in time.* These results illustrate that generalization in AIL is substantially improved through reward deployment in addition to reward learning.

## 4.2 REAL-SIM-REAL NAVIGATION FROM A SINGLE OBSERVATION – **Q2**

Real-world evaluation of AIL is currently challenging. RL-like interaction efficiency renders training in simulation more practical than in the real-world (Tai et al., 2018), but demonstrations must realistically still be from the real-world. Nonetheless, it is imperative to evaluate AIL methods on real-world suboptimal data and hardware since results tend to diverge significantly from ideal settings, synthetic data, and simulation (Orsini et al., 2021; Tsurumine & Matsubara, 2022).

Our hardware experiment evaluates GAIL, IRL-MPC, and MPAIL through Real-to-Sim-to-Real: 1) a *single* partially observable (position and body-centric velocity) trajectory is collected from the real-world subject to sensor noise, 2) the method is trained using interactions from simulation, finally 3) the method is deployed zero-shot to the real-world for evaluation. This experiment uses a small-scale RC car platform with an NVIDIA Jetson Orin NX (Srinivasa et al., 2023). For IRL-MPC, the reward and value is trained through GAIL, which subsequently requires hand-tuning for deployment on MPPI (Triest et al., 2023).

*We remark that the direction of travel cannot be uniquely determined by a single state s due to the partially observable*

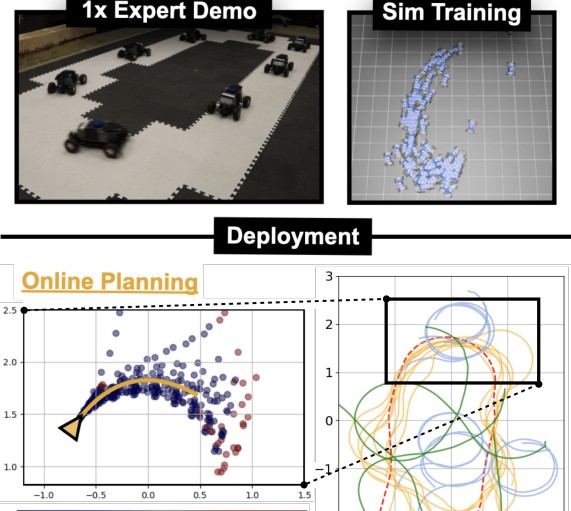

Figure 6: **Real-Sim-Real Experiment. Bottom Left (MPAIL).** Real-time (20 Hz) parallel model rollouts and costing are visualized while the robot navigates through the turn. Current optimal plan for the next 1 second in gold. **Bottom Right.** Trajectories performed by MPAIL, GAIL, and IRL-MPC (see Table 2 for evaluation).

body-centric *velocity*. Only with the state-transition $(s_E, s'_E)$ is it possible to deduce the direction of travel. This property is detailed in Section D.1. Partial observability and state-transitions play critical roles in the recovery of a cost function for this task, presenting a necessary challenge towards practical AIL (Orsini et al., 2021) and scalable Learning-from-Observation (LfO) (Torabi et al., 2019b).

We find that MPAIL is able to qualitatively reproduce the expert trajectory with an average Relative Cross-Track-Error (CTE (Rounsaville et al.)) of 0.17 m while traveling an average of 0.3 m/s slower. In addition, Figure 6 illustrates a key advantage of planning-based AIL. By granting access to the agent's optimization landscape, MPAIL significantly improves on the interpretability of agents trained through observations of ambiguous and complex human data when compared to black-box policies. Note the lower costing of on-track trajectories and final plan.

| | Reward $r(s, s')$ | Policy Optimizer | Deployment |
|---|---|---|---|
| GAIL | $\log(D)$ | PPO (Offline) | Policy |
| AIRL | $\log(\frac{D}{1-D})$ | PPO (Offline) | Policy |
| IRL-MPC | $\log(D)$ | PPO (Offline) | Planner |
| MPAIL | $\log(\frac{D}{1-D})$ | MPPI (Online) | Planner |

Table 1: **Summary of Baselines**. The discriminator is denoted $D := D(s, s')$. Value estimation is performed via GAE-$\lambda$ (Schulman et al., 2018) for all methods.

GAIL does not reliably converge to the expert even in training. During deployment, GAIL's policy consistently veers off-path or collapses into driving in a circle. Various starting configurations were attempted without success. While literature on the evaluation of AIL methods in the real-world are

sparse, we find that AIL policies can be extremely poor performing in the real-world, as corroborated by (Sun et al., 2021). A more detailed discussion is provided in Section D.1.

IRL-MPC acts as a middle-ground between MPAIL and GAIL; the learned reward and value are exactly the same as GAIL's and thus differs by the deployment of the reward through planning. IRL-MPC's improvements over GAIL provides evidence that: (i) model-based planning can grant robustness to a model-free reward and, (ii) despite GAIL's poor performance, the learned reward was still meaningfully discriminative and suggests a failure of the policy to arrive at a solution under the reward. On the other hand, IRL-MPC diverges from MPAIL by mainly learned reward and value. As a result, we find that online policy optimization through $\pi_{\text{MPPI}}$ induces a more competitive adversarial dynamic than offline policy optimization as in actor-critic RL. In this case, the end-to-end inclusion of the planner enables training the reward and value to completion.

Meanwhile, MPAIL's success and IRL-MPC's improvement over GAIL is attributed to model-based planning capabilities. If the robot should find itself away from the expert distribution, the online planner enables the agent to sample back onto the demonstration. In this sense, an MPC-based (or any online-optimizing) agent brings control-theoretic disturbance rejection online. Policies, on the other hand, are far more susceptible to erratic behavior in the real-world due to open-loop action prediction. This becomes especially important when the demonstration data is severely under-defined as in this partially observable setting, which results in ambiguous reward signal in most states in the environment; recall that low discriminator confidence is reflected by low magnitude logit $f_\theta$ (cost) predictions. Real experiment cost values are in the range of $(-0.022, -0.0180)$ whereas cost values in benchmarking runs with synthetic demonstrations are in the range of $(-3, 3)$. On real data, the discriminator also required more frequent updates to provide more reliable signals (see Table 3).

|         | CTE (m) | | |
|---------|-----|------|---------------------|
|         | Max | Mean | Average Speed (m/s) |
| Expert  | -    | -    | 1.0  |
| GAIL    | 1.29 | 0.56 | 0.37 |
| IRL-MPC | 1.28 | 0.37 | 0.30 |
| MPAIL   | 0.76 | 0.17 | 0.70 |

Table 2: **Evaluation of Real Experiment.** Relative Cross-Track Error (CTE) and speed are computed over the best five laps. AIRL is excluded as it does not exhibit meaningful behavior even in simulation.

## 4.3 EFFICIENCY – Q3

As mentioned at the beginning of Section 4, MPAIL does not possess a persistent "policy" as it employs a zeroth-order optimization with bootstrapped value estimates (i.e. infinite-horizon MPPI) to instead resolve these policies online. While we have shown that this *"deconstructed policy"* significantly improves generalization and robustness, concerns regarding interaction efficiency may arise due to the lack of gradient-based value optimization as in actor-critic policy optimization (Schulman et al., 2017b). Here, we perform additional benchmarking experiments to evaluate these hypotheses.

We train GAIL, AIRL, and MPAIL on the navigation task and the cartpole task across varying quantities of expert demonstrations and random seeds as done in (Ho & Ermon, 2016; Kostrikov et al., 2018). While MPAIL uses an approximate prior model for the navigation task, we choose to learn a model during training of the

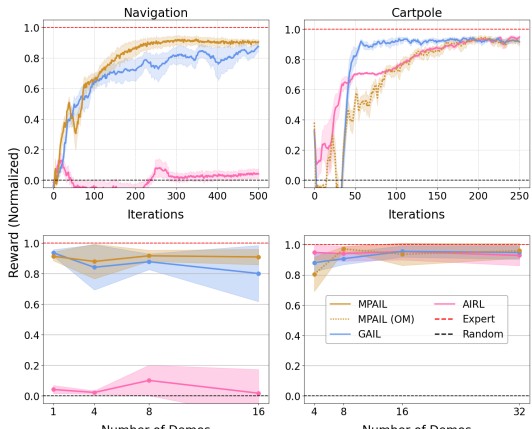

Figure 7: **Benchmarking Results.** Top row rewards are computed across all demonstration quantities and seeds. Bottom row rewards are the average of the final 10 episodes computed across seeds. See Figure 11 for de-aggregated plots.

cartpole task to demonstrate the generality of MPC and support future work on additional tasks. This is represented by the label, *MPAIL (OM)*, indicating that there is a fully **O**nline **M**odel. Implementation details of the learned model can be found in Section D.3.

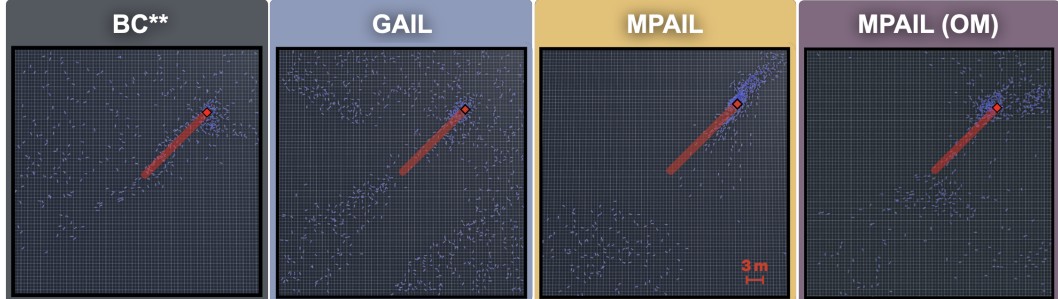

Figure 8: **Out-of-Distribution (OOD) performance evaluation (as in Figure 4) at** $t = 100$ **with MPAIL (OM) and Behavior Cloning (BC) included**. **BC requires access to expert actions and is not an LfO baseline. MPAIL (OM) indicates that a dynamics model is learned online and used for planning in MPAIL. Without a prior model, MPAIL (OM) is limited to the same amount of total information as GAIL; any improvement in OOD generalization over a policy network is purely a result of learning a deconstructed policy for online planning. Details of MPAIL (OM) in Section 4.3 and Section D.3.

On the navigation task, MPAIL reaches optimality in less than half the number of interactions when compared to GAIL. We also observe MPAIL to train more stably than GAIL on this task. AIRL struggles to learn from the multimodal data (see Section 4). In addition to alleviating concerns regarding efficiency, these results support MPAIL's characterization as a model-based algorithm as it is more sample-efficient when provided an (approximate) prior model.

On the cartpole task, expert demonstration data results in optimal-but-sparse state visitation and, equivalently, AIL reward signal. It is likely that, due to online dynamics model learning, MPAIL (OM) requires more exploratory interactions to combat a large local minima induced and reinforced by sparse discriminator reward, model bias, and task dynamics. Similar performance between MPAIL and AIRL may also suggest that GAIL benefits from its inherent reward bias on this task (Kostrikov et al., 2018). Importantly, MPAIL attains comparable performance while maintaining the benefits of model-based planning, such as interpretability, transferability, and robustness.

ADDITIONAL RESULTS

In the Appendix, additional experiments are performed to holistically evaluate MPAIL for the purposes of real-world deployment and robot learning. Wall clock time comparisons, architecture, hyperparameters, model ablations, proofs, de-aggregated benchmark results, and more experiment details and discussion can be found in the appendix. We briefly highlight some key auxiliary results:

**Wall Clock Time.** Our timing evaluations reveal that "inference" and training times of MPAIL can be faster or slower than GAIL (PPO) depending mostly on the number of MPPI iterations per step and the planning horizon $H$. For $H = 10$, MPAIL is about $10\%$ faster than GAIL at 2 iterations. At 5 iterations, the same horizon is about $10\%$ slower. More settings are evaluated and shown in Figure 9.

**MPAIL (OM) — OOD Experiment.** *Does OOD robustness hold when the dynamics model is also learned?* Figure 8 compares the final states of the OOD evaluation in Section 4.1 with MPAIL (OM) included. Note that MPAIL (OM) has access to the same amount of information as the policy-based method. Nevertheless, there is clear improvement in OOD performance by MPAIL (OM) when compared to policy networks. These results crucially suggest that generalizability advantages of the deconstructed policy extends to learned dynamics models in addition to reward and value.

Even in the case of Behavior Cloning (BC), which is directly supervised with access to expert actions, generalization of the policy network appears random and unpredictable. For policy networks (BC and GAIL), there are numerous agents that are initialized near the expert distribution (within red highlight) but remain there. These agents were often merely initialized facing away from the goal, demonstrating that policy networks tend to learn incredibly brittle representations. Most MPAIL

(OM) agents that are seen far away from the goal will continue to slowly arrive at the goal, visually evident by their orientations directed towards the goal. Contrasting with MPAIL, MPAIL (OM) agents tend to follow longer, less optimal paths when strongly OOD. This is likely a result of the learned dynamics being OOD and disrupting planning.

## 5 CONCLUSION

This work adopts an imitation learning setting familiar to humans and animals in which: (1) expert actions are not known, (2) few demonstrations are observed, and (3) the agent infers intent and improves through interaction. While work in Inverse Reinforcement Learning (IRL) and Adversarial Imitation Learning (AIL) from Observation continue to advance these goals, their applications to real-world robots lag behind. To bridge this gap, we observe that connections to model-based planning offers potential towards (a) improved efficiency and transfer, (b) safe and steerable design, and (c) robustness through online optimization. We call this problem setting Planning-from-Observation (PfO). We address PfO by introducing planning-based AIL as a unification of IRL and MPC. Model Predictive Adversarial Imitation Learning (MPAIL) is then introduced as an implementation of planning-based AIL, where a planner is continually improved through cost and value learning.

Towards robot learning applications, we conducted three elucidating evaluations. In Section 4.1, we reveal that reward deployment—not only reward learning as in current AIL—is critical towards generalizable imitation learning. Re-introducing the learned reward online alleviates the burden on policies to generalize and requires the reward, or intent, to generalize instead. In Section 4.2, we see that reward deployment must be met with online optimization for real-world robustness. Especially in partially observable AIL, it may not be sufficient to utilize a planner only during deployment, but the planner should also be included in the learning process to train the reward to completion. From a single partially observable demonstration in real-world navigation, we find that MPAIL is the only successful imitator when compared to policy-based AIL and IRL-MPC. MPAIL also employs representations (e.g. model, reward, value) which grant direct access to the agent's optimization landscape and thus decision-making process—a powerful prerequisite towards safe and interpretable robot learning. Finally, in Section 4.3, interaction efficiency benchmarks favorable to MPAIL with an approximate prior model address concerns regarding MPPI's zeroth-order policy optimization and empirically supports MPAIL's characterization as a model-based algorithm.

MPAIL is derived from, and naturally admits, abstractions from Model Predictive Control, model-based RL, and imitation learning. Thus, its open-source implementation aims to reflect this and offers common ground for instantiating the many possible extensions to adjacent work through these connections: off-policy value estimation for improved sample efficiency or offline learning (Kostrikov et al., 2018), policy-like proposal distributions and latent dynamics for scaling MPPI to higher dimensional spaces (Hansen et al., 2024; Zhu et al., 2025), model-free and model-based reward blending for alleviating model bias (Bhardwaj et al., 2021), diffusion-inspired MPPI for improved online optimization (Xue et al., 2024), and much more (Han et al., 2026). We envision that this work can provide a theoretically and empirically justified foundation for future work at the intersection of MPC, RL, and Imitation Learning.

## REPRODUCIBILITY

Included in the supplementary material is the code for MPAIL, developed with the intention to be released as a standalone package. Configuration files and hyperparameters for experiments are almost entirely "flat" (viewable in the code) and are as presented in Table 3. Simulation environment is built with Isaac Lab (Mittal et al., 2023) and one of its extensions, Wheeled Lab (Han et al., 2025). Instructions for installation can be found in their respective GitHub repositories. Hardware deployment code is also included for the (open-source) platform, MuSHR (Srinivasa et al., 2023); an example script detailing how to save and load an MPAIL planner onto the hardware can be found in the supplementary as well. More details explaning source code organization, as well as experiment videos, can be found in the README.md. We welcome efforts towards reproducibility and encourage practitioners to correspond regarding difficulties.

## ACKNOWLEDGMENTS

TH is supported by the NSF GRFP under Grant No. DGE 2140004.

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

Brian D Ziebart, Andrew Maas, J Andrew Bagnell, and Anind K Dey. Maximum Entropy Inverse Reinforcement Learning.

# APPENDIX

## A  INFINITE HORIZON MODEL PREDICTIVE PATH INTEGRAL

In this section we present the full algorithm in detail, including MPPI as described in (Williams et al., 2017). Modifications to "conventional" MPPI for MPAIL are highlighted in blue. Where applicable, $(\mathbf{x})_i$ indicates the $i$th entry of $\mathbf{x}$ (in its first dimension, if $\mathbf{x}$ is a tensor).

---

**Algorithm 2** MPPI

---

**Require:**
  Number of trajectories to sample $N$;
  Planning horizon $H$;
  Number of optimization iterations $J$
  Fixed action sampling variance $\Sigma$;
  Previous optimal plan $\mathbf{a}_{t-1}^* = \{(\mathbf{a}_{t-1}^*)_{t'}\}_{t'=0}^H$;
  Current state $s_t$;
  Dynamics model $f_\psi(s, a)$
  Costs $c_\theta(s, s')$
  Value $V_\phi(s)$

1: **Procedure** MPPI$(s_t, \mathbf{a}_{t-1}^*)$

2:  $(\mathbf{a}_t)_i^0 \leftarrow (\mathbf{a}_{t-1}^*)_{i+1}$      ▷ Roll previous plan one timestep forward
3:  $(\mathbf{a}_t)_H^0 \leftarrow 0$        ▷ Set sampling mean to 0 for last timestep
4:  **for** $j \leftarrow 0$ **to** $J - 1$ **do**
5:   **for** $k \leftarrow 0$ **to** $N - 1$ **do**  ▷ Model rollouts and costing (parallelized)
6:    $\tilde{s}_0^k \leftarrow s_t$
7:    **for** $t' \leftarrow 0$ **to** $H - 1$ **do**
8:     $a_{t'}^k \sim \mathcal{N}((\mathbf{a}_t)_{t'}, \Sigma)$      ▷ Sample action at predicted state
9:     $\tilde{s}_{t'+1}^k \leftarrow f_\psi(\tilde{s}_{t'}^k, a_{t'}^k)$        ▷ Predict next state
10:     $c_{t'}^k \leftarrow c_\theta(s_{t'}^k, s_{t'+1}^k)$      ▷ Compute state-transition costs
11:    **end for**
12:    $\mathcal{C}(\tau_k) \leftarrow -\eta^H V_\phi(\tilde{s}_H^k) + \sum_{t'=0}^{H-1} \eta^{t'} c_{t'}^k$     ▷ Total trajectory cost
13:   **end for**
14:   $\beta \leftarrow \min_k[\mathcal{C}(\tau_k)]$
15:   $\mathcal{Z} \leftarrow \sum_{k=1}^n \exp{-\frac{1}{\lambda}\mathcal{C}(\tau_k)}$
16:   **for** $k \leftarrow 0$ **to** $N - 1$ **do**   ▷ Weight using exponential negative cost
17:    $w(\tau_k) \leftarrow \frac{1}{\mathcal{Z}} \exp{-\frac{1}{\lambda}\mathcal{C}(\tau_k)}$
18:   **end for**
19:   **for** $t' \leftarrow 0$ **to** $H - 1$ **do**  ▷ Optimal plan from weighted-average actions
20:    $(\mathbf{a}_t^j)_{t'} \leftarrow \sum_{k=0}^{N-1} w(\tau_k) a_{t'}^k$
21:   **end for**
22:  **end for**
23:  **for** $i \leftarrow 0$ **to** $|\mathcal{A}|$ **do** ▷ Compute optimized standard deviations for policy
24:   $(\boldsymbol{\sigma}_t)_i \leftarrow \sqrt{\sum_{k=0}^{N-1} w(\tau_k)[((\mathbf{a}_t^J)_0)_i - (a_0^k)_i]^2}$
25:  **end for**
26:  **return** $\mathbf{a}_t^J, \boldsymbol{\sigma}_t$
27: **End Procedure**

---

## B  PROOFS

In Section 3, we introduce the replacement of the entropy loss in Equation (2) with a KL divergence loss. This replacement allows the MPPI planner, in place of a policy, to solve the required forward RL problem. Integrated with the AIL objective in Equation (2), we further show that this allows MPAIL to correctly recover the expert state occupancy distribution $\rho_E(s, s')$. In this section, we prove both of these claims.

---

**Algorithm 3** $\pi_{\text{MPPI}}$

---

**Require:**
    Reward $r_\theta := -c_\theta$;
    Value $V_\phi$;
    $\text{MPPI}(s, \mathbf{a}) = \text{MPPI}(s, \mathbf{a}; N, H, J, \Sigma, f_\psi, c_\theta, V_\phi)$ (Algorithm 2);
    $T$ length of episode
1: $\mathbf{a}_0^* \leftarrow 0$                                         ▷ Initialize optimal plan
2: $\mathcal{B} \leftarrow \{\}$
3: **for** $t \leftarrow 1$ **to** $T$ **do**
4:     $s_t \sim \mathcal{T}(\cdot | s_{t-1}, a_{t-1})$                        ▷ Step and perceive environment
5:     $\mathbf{a}_t^*, \boldsymbol{\sigma}_t \leftarrow \text{MPPI}(s_t, \mathbf{a}_{t-1}^*)$
6:     **if** Train **then**
7:         $a_t \sim \mathcal{N}((\mathbf{a}_t^*)_0, I\boldsymbol{\sigma}_t)$
8:     **else if** Deploy **then**
9:         $a_t \leftarrow (\mathbf{a}_t^*)_0$
10:    **end if**
11:    $r_t \leftarrow r_\theta(s_t, s_{t+1})$                              ▷ Reward from discriminator
12:    $\mathcal{B} \leftarrow \mathcal{B} \cup (s_t, a_t, r_t, s_{t+1})$
13: **end for**
14: **return** $\mathcal{B}$

---

## B.1    MPPI AS A POLICY

In this section we justify claims regarding MPPI in the forward RL problem. For completeness, we also verify that known results remain consistent with our state-only restriction.

**Proposition B.1.1.** *The closed form solution of Equation* (3),

$$\min_{\pi \in \Pi} \mathbb{E}_\pi[c(s, s')] + \beta \, \mathbb{KL}(\pi \,||\, \overline{\pi}), \tag{3}$$

*is*

$$\pi^*(a|s) \propto \overline{\pi}(a|s) e^{\frac{-1}{\beta} \overline{c}(s,a)} \quad \text{where} \quad \overline{c}(s,a) = \sum_{s' \in \mathcal{S}} \mathcal{T}(S_{t+1} = s' | S_t = s) c(s, s') \tag{7}$$

*Proof.* We begin by noting that

$$\mathbb{E}_\pi[c(s, s')|S_t = s] = \sum_{a \in \mathcal{A}} \pi(a|s) \overline{c}(s, a) \tag{8}$$

where the weighted cost $\overline{c}(s, a)$ is defined as

$$\overline{c}(s, a) := \sum_{s' \in \mathcal{S}} \mathcal{T}(S_{t+1} = s' | S_t = s) c(s, s') \tag{9}$$

Then for a fixed state $s \in \mathcal{S}$, noting that the policy is normalized over actions $\sum_{a \in \mathcal{A}} \pi(a|s) = 1$, we may form the Lagrangian with respect to the objective in Equation (3) as:

$$\mathcal{L}(\pi, \beta, \lambda) = \sum_{a \in \mathcal{A}} \pi(a|s) \overline{c}(s, a) \ + \ \beta \sum_{a \in \mathcal{A}} \pi(a|s) \log \frac{\pi(a|s)}{\overline{\pi}(a|s)} \ + \ \lambda \sum_{a \in \mathcal{A}} \pi(a|s) \ - 1 \tag{10}$$

Taking the partial derivative with respect to $\pi(a|s)$ and setting to 0 we have

$$\frac{\partial \mathcal{L}}{\partial \pi(a|s)} = \overline{c}(s, a) + \beta \log \frac{\pi(a|s)}{\overline{\pi}(a|s)} + 1 + \lambda = 0 \tag{11}$$

Finally,

$$\beta \log \frac{\pi(a|s)}{\overline{\pi}(a|s)} = -\overline{c}(s, a) - 1 - \lambda \tag{12}$$

$$\pi(a|s) \propto \overline{\pi}(a|s) e^{-\frac{1}{\beta}(\overline{c}(s,a) + 1 + \lambda)} \tag{13}$$

$$\pi(a|s) \propto \overline{\pi}(a|s) e^{-\frac{1}{\beta} \overline{c}(s,a)} \qquad \qquad \square$$

**Remark B.1.2.** *Given a uniform policy prior, the KL Objective in Equation* (3),

$$\min_{\pi \in \Pi} \mathbb{E}_\pi[c(s, s')] + \beta \, \mathbb{KL}(\pi \, || \, \overline{\pi}), \tag{3}$$

*is equivalent to the Entropy Objective,*

$$\min_{\pi \in \Pi} \mathbb{E}_\pi[c(s, s')] - \lambda \, \mathbb{H}(\pi). \tag{14}$$

*Proof.* In order to prove this, it suffices to note that minimizing KL is equivalent to maximizing entropy:

$$\mathbb{KL}(\pi || \overline{\pi}) = \sum_{s \in \mathcal{S}} d^\pi(s) \sum_{a \in \mathcal{A}} \pi(a|s) \log \frac{\pi(a|s)}{\overline{\pi}(a|s)} \tag{15}$$

$$= \sum_{s \in \mathcal{S}} d^\pi(s) \left[ -\mathbb{H}(\pi(\cdot|s)) - \sum_{a \in \mathcal{A}} \pi(a|s) \log \overline{\pi}(a|s) \right] \tag{16}$$

$$= -\sum_{s \in \mathcal{S}} d^\pi(s) \mathbb{H}(\pi(\cdot|s)) - \sum_{s \in \mathcal{S}} \sum_{a \in \mathcal{A}} \pi(a|s) \log \overline{\pi}(a|s) \tag{17}$$

$$= -\mathbb{H}(\pi) - \sum_{s \in \mathcal{S}} \log k_s \tag{18}$$

where $k_s = \overline{\pi}(a|s)$ for any $a \in \mathcal{A}$. Note that the sum on the left collapses by definition and the inner sum on the right collapses since the probability of taking an action in any given state is 1. Finally, since all the $k_s$ are constant, the second term on the right hand side is constant. Since both objectives differ by a constant, minimizing the KL is equivalent to maximizing the Entropy given a uniform policy prior. $\square$

**Proposition B.1.3.** *Provided the MDP is uniformly ergodic, the MPPI objective in Equation* (4),

$$\min_{\pi \in \Pi} \mathbb{E}_{\tau \sim \pi} \left[ C(\tau) + \beta \, \mathbb{KL}(\pi(\tau) \, || \, \overline{\pi}(\tau)) \right] \tag{4}$$

*is equivalent to the RL objective in Equation* (3),

$$\min_{\pi \in \Pi} \mathbb{E}_\pi[c(s, s')] + \beta \, \mathbb{KL}(\pi \, || \, \overline{\pi}). \tag{3}$$

*Proof.* Before continuing, we verify that infinite horizon MPPI indeed predicts an infinite horizon estimate of the return. For simplicity, we momentarily revert to a reward only formulation, replacing the cost $c_\theta(s, s')$ with the reward $R_\theta(s, s')$ and the control discount $\eta$ with $\gamma$. We proceed by expanding the return,

$$\mathbb{E}_{\tau \sim \pi}[R(\tau)] = \mathbb{E}_{\tau \sim \pi}[\gamma^H V_\phi(s_H) + \sum_{t=1}^{H-1} \gamma^t R(s_t, s_{t+1})] \tag{19}$$

$$= \mathbb{E}_{\tau \sim \pi}[\mathbb{E}_\pi[\sum_{t=H}^{\infty} \gamma^t R(s_t, s_{t+1}) | S_H = s_H] + \sum_{t=1}^{H-1} \gamma^t R(s_t, s_{t+1})] \tag{20}$$

$$= \mathbb{E}_{\tau \sim \pi}[\sum_{t=H}^{\infty} \gamma^t R(s_t, s_{t+1}) + \sum_{t=1}^{H-1} \gamma^t R(s_t, s_{t+1})] \tag{21}$$

$$= \mathbb{E}_{\tau \sim \pi}[\sum_{t=1}^{\infty} \gamma^t R(s_t, s_{t+1})] \tag{22}$$

where we have made use of the definition of a value function $V_\phi$ (Equations 18 to 19) and the tower property of expectation (Equations 19 to 20).

Let $f(s, s') = c(s, s') + \beta \mathbb{KL}(\pi(\cdot|s) || \overline{\pi}(\cdot|s))$. For either objective to be valid the cost and KL Divergence would have to be bounded. Thus, we may safely assume that $f$ is uniformly bounded $||f||_\infty \leq K$. Let $\delta_t = \mathbb{E}_{s_t, s_{t+1} \sim d^t}[f(s_t, s_{t+1})] - \mathbb{E}_{s, s' \sim d^\pi}[f(s, s')]$ be the error between estimates of the objective.

Since the MDP is uniformly ergodic, we may bound the rate of convergence of the state distribution at a time $t$, $d^t$ to the stationary distribution $d^\pi$

$$\exists \lambda \in (0, 1), M \in \mathbb{N} \text{ s.t } ||d^t - d^\pi||_{\text{TV}} \leq M\lambda^t \tag{23}$$

where $|| \cdot ||_{\text{TV}}$ is the total variation metric.

Continuing by bounding error $\delta_t$,

$$|\delta_t| \leq ||f||_\infty ||d^t - d^\pi||_{\text{TV}} \leq KM\lambda^t \tag{24}$$

We then have that $|\sum_{t=0}^\infty \eta^t \delta_t| \leq KM \sum_{t=0}^\infty (\eta\lambda)^t = \frac{KM}{1-\eta\lambda} = C < \infty$.

We may now begin working with the MPPI Objective in Equation (4)

$$\mathbb{E}_{\tau \sim \pi}[C(\tau) + \beta \mathbb{KL}(\pi(\tau) || \overline{\pi}(\tau))] \tag{25}$$

$$= \sum_{t=0}^\infty \eta^t \mathbb{E}_{s_t, s_{t+1} \sim d^t}[f(s_t, s_{t+1})] \tag{26}$$

$$= \sum_{t=0}^\infty \eta^t [\mathbb{E}_{s, s' \sim d^\pi}[f(s, s')] + \delta_t] \tag{27}$$

$$= \frac{1}{1-\eta} \mathbb{E}_{s, s' \sim d^\pi}[f(s, s')] + \sum_{t=0}^\infty \eta^t \delta_t \tag{28}$$

Note that the MPPI objective and Entropy Regularized RL objective differ by scaling and a bounded additive constant, independent of $\pi$. Thus, minimizing both objectives are equivalent. $\square$

## B.2 MPAIL as an Adversarial Imitation Learning Algorithm

In this section, we integrate findings from Section B.1 with the AIL objective to theoretically validate MPAIL as an AIL algorithm. Specifically, we observe that at optimality, we recover the log expert-policy transition density ratio, which in turns yields a maximum entropy policy on state-transitions. We then discuss the identifiability limits imposed by observing only $(s, s')$ rather than $(s, a, s')$. Throughout this section we make use of the state-transition occupancy measure, defined as $\rho_\pi$ : $\mathcal{S} \times \mathcal{S} \to \mathbb{R}$ where $\rho_\pi(s, s') = \sum_{t=1}^\infty \gamma^t \mathcal{T}(S_{t+1} = s', S_t = s \,|\, \pi)$ as in (Torabi et al., 2019a).

**Proposition B.2.1.** *The optimal reward is*

$$f_\theta^*(s, s') = \log\left(\frac{\rho_E(s, s')}{\rho_\pi(s, s')}\right) \tag{29}$$

*Proof.* Note that the optimal discriminator is achieved when $D^*(s, s') = \frac{\rho_E(s,s')}{\rho_E(s,s')+\rho_\pi(s,s')}$ as used in (Ghasemipour et al., 2020) and shown in (Goodfellow et al., 2014) Section 4 Proposition 1.

$$f_\theta^*(s, s') = \log(D^*(s, s')) - \log(1 - D^*(s, s')) \tag{30}$$

$$= \log\left(\frac{\rho_E(s, s')}{\rho_E(s, s') + \rho_\pi(s, s')}\right) - \log\left(\frac{\rho_\pi(s, s')}{\rho_E(s, s') + \rho_\pi(s, s')}\right) \tag{31}$$

$$= \log\left(\frac{\rho_E(s, s')}{\rho_\pi(s, s')}\right) \qquad\qquad \square$$

This shows that, by setting $r(s, s') = f_\theta(s, s')$, the recovered reward function is the log-ratio of state-transition occupancy measure from the expert to the policy.

**Lemma B.2.2.** *MPAIL minimizes a regularized KL divergence between the policy's state-transition occupancy measure and the expert's.*

*Proof.* Recall from Proposition B.1.1 that while solving for the RL objective, MPPI finds a policy of the form

$$\pi(a|s) \propto \overline{\pi}(a|s)e^{-\frac{1}{\beta}\overline{c}(s,a)} \tag{32}$$

Applying Proposition B.2.1, we plug $c(s, s') = -f_\theta^*(s, s')$ into Equation (9) to obtain

$$\pi^*(s, a) \propto \overline{\pi}(s, a) \exp\left(-\frac{1}{\beta} \sum_{s' \in \mathcal{S}} \mathcal{T}(s'|s)[\log \rho_\pi - \log \rho_E)]\right) \tag{33}$$

Note that when the policy distribution $\rho_\pi$ matches the expert distribution $\rho_E$ the exponential term collapses. Thus, when the occupancy measures match, the policy updates cease to have an effect and the optimization attains a fixed point.

In fact, we may note that for any fixed state $s \in \mathcal{S}$, the cost accumulated by the policy is

$$\sum_{s' \in \mathcal{S}} \rho_\pi(s, s')c^*(s, s') = \sum_{s' \in \mathcal{S}} \rho_\pi(s, s') \log\left(\frac{\rho_\pi(s, s')}{\rho_E(s, s')}\right) = \mathbb{KL}(\rho_\pi(s, \cdot)||\rho_E(s, \cdot)) \tag{34}$$

Finally, the MPPI objective can be written as

$$\min_{\pi \in \Pi} \mathbb{KL}(\rho_\pi||\rho_E) + \beta \, \mathbb{KL}(\pi \,||\, \overline{\pi}) \tag{35}$$

showing that the MPAIL procedure minimizes the entropy regularized KL Divergence between state-transition occupancy measures. In this sense, we have shown that MPAIL can be indeed classified as an AIL algorithm which seeks to match the expert's occupancy measure through an MPPI Policy.

$\square$

**Remark B.2.3.** *On Identifiability.* A question naturally arises about the limitations that being state-only imposes. If state transitions are deterministic and invertible, observing $(s, s')$ is the same as observing the unique action $a$ that caused it. Then $r(s, s') = r(s, a)$ and by the 1-1 correspondence of policies with state-action occupancy measures (Ho & Ermon, 2016), the recovered policy becomes unique.

In general, this assumption has varying degrees of accuracy. When transitions are many to one or stochastic, multiple actions can produce the same transition $(s, s')$. Then $\rho_\pi(s, s') = \rho_\pi(s) \sum_{a \in \mathcal{A}} \pi(a|s)\mathcal{T}(s'|s, a)$ becomes a mixture over actions which induces a range of respective policies. For instance, if $\mathcal{T}(s'|s, a_1) = \mathcal{T}(s'|s, a_2)$ for actions $a_1, a_2$, the expert could perform either action and a state-transition based reward would not distinguish between them. Nonetheless, IRL is already an ill-posed problem due to the many to one relationships between policies, rewards, and demonstrations. Though the ambiguity is exacerbated by lack of demonstrated actions, it is still inherent to the problem.

## C  IMPLEMENTATION

In this section we provide further details about the algorithm implementation. Some features incorporated here are deemed well-known (i.e. spectral normalization) or not rigorously studied for statistical significance but included for completeness and transparency.

## C.1 REGULARIZATION

**Spectral Normalization.** As often found in GAN and AIL surveys (Orsini et al., 2021; Miyato et al., 2018), we corroborate that applying spectral normalization to the discriminator architecture appeared to have improved MPAIL training stability and performance. Application of spectral normalization to the value network did not appear to make a noticeable difference.

**L2 Weight Regularization.** Some experimentation was done with L2 weight regularization, but it was ultimately *not used for any simulation results*. Instead, usage of the weight regularization for the real experiment (Section 4.2) appeared to help stabilize training and allow for more reliable model selection and deployment.

## C.2 HYPERPARAMETERS

Fundamentally derived from AIL and MPPI, we can etymologically partition hyperparameters into those induced by AIL (orange) and by MPPI (blue). Remaining non-highlighted parameters for this work are introduced and discussed below.

**Temperature Decay.** As noted in Section 3, we found that an initial temperature with a gradual decay (down to a minimum) was helpful in preventing early and unrecoverable collapse. The intuition for this decision is similar to that of decaying policy noise injection in many popular RL frameworks (Lillicrap et al., 2019; Hansen et al., 2024), since the temperature is directly related to the variance of the optimized gaussian distribution. This component remains under investigation as its usage is not always necessary for meaningful convergence, but it is perhaps practically useful as it alleviates temperature tuning labor.

**Value-to-Discriminator Update Ratio.** Common to existing AIL (and GAN) implementations, MPAIL benefits from a balancing of generator and discriminator updates. Note that, like

| Hyperparameter | Value |
|---|---|
| Disc. optimizer ($\theta$) | Adam ($\beta_1 = 0.5$, $\beta_2 = 0.999$) |
| Disc. learning rate | 1e-4 |
| Disc. hidden width | 32 |
| Disc. hidden layers | 2 |
| Disc. L2 coefficient | 0 (sim), 0.001 (real) |
| Value optimizer ($\phi$) | Adam ($\beta_1 = 0.9$, $\beta_2 = 0.999$) |
| Value learning rate | 1e-3 |
| Value hidden width | 32 |
| Value hidden layers | 2 |
| Value loss clip | 0.2 |
| Discount ($\gamma$) | 0.99 |
| Generalized return ($\lambda$) | 0.95 |
| Value max grad norm | 1.0 |
| Mini batches | 3 |
| Epochs | 3 |
| Trajectories ($N$) | 512 |
| Planning horizon ($H$) | 10 |
| Iterations ($J$) | 5 |
| Sampling variance ($\Sigma$) | $diag([0.3 \ldots])$ |
| Initial temperature ($\lambda_0$) | 1.0 |
| Markup/Discount ($\eta$) | 1.01 |
| Temp. decay rate | 0.01 |
| Minimum temp. | 1e-5 |
| Value:Disc update ratio | 3:1 (sim), 1:1 (real) |

Table 3: **MPAIL Hyperparameters**. Used across all experiments unless specified otherwise.

GANs, AIL tends to oscillate aggressively throughout training (Luo et al., 2024). As MPAIL does not enforce a constrained policy update each epoch (as TRPO does (Schulman et al., 2017a)), the policy is exposed more directly to the discriminator's oscillations which can further hinder on-policy value estimation. A further converged value function is also theoretically more stationary from the perspective of $\pi_{\text{MPPI}}$ as infinite-horizon MPPI.

**Markup.** A notable quirk discovered during implementation is the relationship between costs and rewards. While the two concepts are generally regarded as dual (with negation), it is worthwhile noting that discounting is not closed under negation. Meaning, it is not correct to apply the same discount factor to the costs as they are done to the summation of rewards in the return $G_t$. Consider the reward with a discount applied $r_1 = \gamma \, r(s, s')$ and the reward of a cost with a discount applied $r_2 = -\gamma \, c(s, s')$. Observe that for $\gamma < 1$, $r_1$ decreases while $r_2$ increases. Thus, when using costs, the $H$-step factor in the MPPI horizon, $\eta$, should not decrease over $t'$. In fact, when applying $\eta < 1$, we found that MPAIL does not meaningfully converge ever on the navigation task. (Geldenbott & Leung, 2024) names the usage of $\eta > 1$ as a markup. In our case, we apply a similar empirical factor such that $\eta := 1/\gamma > 1$. While we suspect a more rigorous relationship between $\eta$ and $\gamma$, we leave its derivation for future work. However, we remark that a reward-only variant of MPPI which precludes these relationships is equally possible as done in (Hansen et al., 2022). Costs are maintained in this work due to wider familiarity in practice (Han et al., 2024a; Williams et al., 2017; Morgan et al., 2021; Finn et al., 2016).

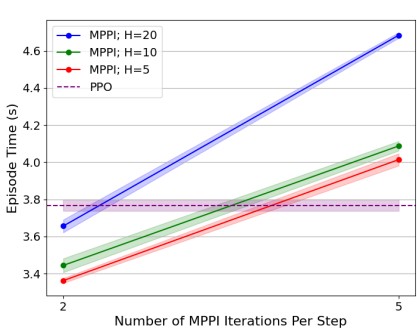

Figure 9: **Comparing "Inference" Times for Navigation Task.** Time taken to complete one episode of 100 timesteps with 64 parallel environments across varying horizon lengths and MPPI optimization iterations. PPO (in policy-based AIL) is used as implemented in the RSL library (Rudin et al., 2022). All training runs in this work are performed on an NVIDIA RTX 4090 GPU. Isaac Lab is chosen as our benchmarking and simulation environment due to its parallelization and robot learning extensions (Mittal et al., 2023; Han et al., 2025).

```
MPAILPolicy initialized. Total number of params: 3779
Dynamics: 0
Sampling: 0
Cost: 3778
Temperature: 1
MPAILPolicy(
  (dynamics): KinematicBicycleModel()
  (costs): TDCost(
    (ss_cost): GAIfOCost(
      (reward): Sequential(
        (0): Linear(in_features=24, out_features=32,
            bias=True)
        (1): LeakyReLU(negative_slope=0.01)
        (2): Linear(in_features=32, out_features=32,
            bias=True)
        (3): LeakyReLU(negative_slope=0.01)
        (4): Linear(in_features=32, out_features=1,
            bias=True)
      )
    )
    (ts_cost): CostToGo(
      (value): Sequential(
        (0): Linear(in_features=12, out_features=32,
            bias=True)
        (1): ReLU()
        (2): Linear(in_features=32, out_features=32,
            bias=True)
        (3): ReLU()
        (4): Linear(in_features=32, out_features=1,
            bias=True)
      )
    )
  )
  (sampling): DeltaSampling()
)
```

Figure 10: **PyTorch (Paszke et al., 2019) Model Architecture from Train Log.** MPAIL readily admits other well-studied components of the model-based planning framework (e.g. sampling, dynamics) (Vlahov et al., 2025; 2024). This work focuses on costing from demonstration.

### C.3 COMPUTATION

MPAIL is crucially implemented to be parallelized across environments in addition to trajectory optimization. In other words, in a single environment step, each parallel environment independently performs parallelized sampling, rollouts, and costing entirely on GPU without CPU multithreading. MPPI also allows for customize-able computational budget, similar to (Hansen et al., 2024) (see Figure 9). For the navigation and cartpole tasks, we find that online trajectory optimization implemented this way induces little impact on training times. In exchange, MPPI can be more space intensive due to model rollouts having space complexity of $\mathcal{O}(HN|\mathcal{S}|)$ per agent. On the navigation task benchmark settings, this is an additional 245 kB per agent or 15.7 MB in total for 64 environments. Figure 9 shows benchmarks on training times that demonstrate comparable times to PPO's policy inference. Overall, training runs for Section 4.3 between MPAIL and GAIL on the navigation task are comparable at about 45 minutes each for 500 iterations.

## D EXPERIMENTAL DETAILS

### D.1 REAL-SIM-REAL NAVIGATION

**Setup Details.** Before continuing with the discussion of our results, we provide further details about the setup of the experiment. The platform itself is an open-source MuSHR platform as detailed in (Srinivasa et al., 2023). Notably, the compute has been replaced with an NVIDIA Jetson Orin NX as mentioned in Section 4.2. Poses (position, orientation; $[x \quad y \quad z \quad r \quad p \quad y]$) are provided by a motion-capture system at a rate of 20 Hz. Velocities are body-centric as estimated by onboard wheel encoders ($\mathbf{v} = v_x \mathbf{b}_1 + v_y \mathbf{b}_2 + v_z \mathbf{b}_3$, such that $\mathbf{b}_1$ points forward, $\mathbf{b}_2$ points left, and $\mathbf{b}_3$ completes the right-hand frame; basis vectors are rigidly attached to the vehicle (Han et al., 2024a)). Note that the vehicle is operated without slipping nor reversing such that $v_y \approx v_z \approx 0$ and $v_x > 0$ (Han

et al., 2024b). The recorded states used for the expert demonstration data is 240 timesteps long. Altogether, the data can be written as $s_E \in \{(x_t, y_t, z_t, v_{x,t}, v_{y,t}, v_{z,t})\}_{t=1}^{240}$.

A remark: GAIL for this task is necessarily implemented with "asymmetry" between actor and reward. Since, the discriminator must receive as input expert observations $s_E$ while the agent is provided $(r, p, y)$ in addition to observations in $s_E$. In theory, there should be no conflict with the IRLfO (Equation (1)) formulation as this remains a valid reward but on a subset of the state.

**Additional Discussion of Results**.

To understand the role of partial observability in the experiment design, consider a simplified hand-designed cost using the partially observable expert data $c(s, s'|s_E, s'_E)$. A reference vector can be computed through the difference of positions between $s'$ and $s$ then scaled by the demonstrated velocities: $c(s, s'|s_E, s'_E) := \|^{\mathcal{I}}\mathbf{v}(s, s') - v_{xE}[(x'_E - x_E)\mathbf{e}_1 + (y'_E - y_E)\mathbf{e}_2]\|_2$ where $\mathbf{e}_i$ are global basis vectors for global frame $\mathcal{I}$ and $^{\mathcal{I}}\mathbf{v}(s, s')$ is the robot velocity in $\mathcal{I}$. Of course, this example assumes the ability to correctly choose the corresponding $(s_E, s'_E)$ pair for input $(s, s')$ out of the entirety of the expert dataset $d^E$. It should be clear that partial observability and state-transitions play critical roles in the recovery of this non-trivial cost function. This experiment presents a necessary challenge towards practical AIL (Orsini et al., 2021) and scalable Learning-from-Observation (LfO) (Torabi et al., 2019b).

IRL-MPC was evaluated across three ablations: (a) reward-only, (b) value-only, and (c) reward-and-value. The results in Figure 6 reflect the performance of (a) reward-only. The other implementations were distinctly worse than (a) and frequently devolved into turning in circles much like GAIL.

In both cases of GAIL and MPAIL, we find that the agents occasionally travel counter-clockwise (where the expert travels clockwise) during training, suggesting that $(s_E, s'_E)$ appears close to $(s'_E, s_E)$ through the discriminator. As the data is collected through real hardware, it is suspected that state estimation noise introduces blurring between states that are separated by only 50 ms. GAIL is otherwise known to perform poorly in the existence of multi-modal data (Li et al., 2017). This is further corroborated by its unstable performance on the navigation benchmark. And, to the best of our knowledge, similar Real-Sim-Real applications of AIL appear sparse if existent at all. Adjacent works which use real demonstration data but train in real include (Tsurumine & Matsubara, 2022; Sun et al., 2021). Even while training in real, GAIL's performance drops signficantly ($90\% \rightarrow 20\%$) when presented with imperfect demonstrations for even straightforward tasks like reaching (Tsurumine & Matsubara, 2022; Sun et al., 2021). These observations might suggest why the GAIL discriminator is unable to learn meaningfully in simulation and produces a poor policy.

## D.2 SIMULATED NAVIGATION TASK DETAILS

**Reward and Data.** The exact form of the reward used for training PPO and for metrics is given by

$$r(s) := \sqrt{10^2 + 10^2} - \sqrt{(x - 10)^2 + (y - 10)^2}.$$

Figure 3 visualizes four demonstrations from the converged PPO "expert" policy. Additional demonstrations are generated by playing more environments from this policy for one episode such that each demonstration is distinct. Each episode is 100 timesteps long, where each timestep is 0.1 seconds.

**OOD Experiment.** OOD Energy in Figure 5 is computed as described by Liu et al. (2020). Namely, with respect to the expert data $d^E$, we fit a reference distribution using $\tilde{P}_E = \mathcal{N}(\bar{\mu}_E, \bar{\Sigma}_E)$. Then, the OOD energy is given by $E(s; p_E) = \log p_E(s)$. Some limitations of this procedure can be observed given that ID points for GAIL do not receive as much reward as one might expect. However, this remains reasonable considering that the GAIL policy may forget ID behavior, which can be seen in Figure 4 by agents clearly ID remaining static throughout the episode. Future work might better explore quantifying OOD towards measuring AIL generalization through direct usage of the discriminator.

## D.3 PREDICTIVE MODEL LEARNING TOWARDS GENERALIZABLE MPAIL

For tasks beyond navigation (see also Section D.5 for the Ant environment), planning rollouts were generated from a deterministic dynamics model $f_\psi(s, a)$ learned entirely online. The dynamics

model was trained to minimize the mean squared error between the predicted and observed $s_{t+1}$, given $s_t$ and $a_t$. The loss being optimized can be written as:

$$\hat{s}_{i+1} = f_\psi(s_i, a_i), \quad L = \frac{1}{H_B} \sum_{s,a \in B} (s_{i+1} - \hat{s}_{i+1})^T (s_{i+1} - \hat{s}_{i+1}) \tag{36}$$

with model parameters $\psi$, transition buffer $B$, and a mini-batch of size $H_B$ sampled from $B$. If used, the update for the model occurs after line 4 in Algorithm 1.

**Training augmentations.** Several training augmentations were made to improve model accuracy and stability. A transition replay buffer, which stored transitions from multiple episodes, was used to train the dynamics model for multiple epochs during each MPAIL training iteration. After each episode, the buffer was updated by randomly replacing old transitions with those from the latest episode. This off-policy buffer helped stabilize training and

| Dynamics Model Hyperparameter | Value |
|---|---|
| Optimizer | Adam($\beta_1 = 0.9, \beta_2 = 0.999$) |
| Learning rate | 1e-3 |
| LR decay rate | 0.9 |
| LR decay frequency (ep.) | 25 (Ant), 15 (Cartpole) |
| Min. LR | 1e-6 |
| Hidden width | 256 (Ant), 64 (Cartpole) |
| Hidden layers | 3 |

Table 4: **Dynamics Learning Hyperparameters.**

prevent overfitting when training using multiple epochs. Furthermore, applying a step-based learning rate decay improved convergence speed. Dynamics model-specific hyperparameters are listed in Table 4.

### D.4 ABLATIONS

Figure 12 shows the results of an ablation study, investigating the effect of including costs or values in the MPAIL formulation. We find that including both is necessary for reasonable behavior across varying horizon lengths. We observe that value-only planning can quickly improve but is highly unstable and is unreliable as a generator. As expected, cost-only planning performs progressively better at longer planning horizons. However, should the agent find itself off-distribution, it is not able to return to the distribution until it randomly samples back in, which may potentially never occur. For instance, many agents which are initialized facing the opposite direction drive randomly without ever returning to the distribution. Without a value function guiding the agent, the discriminator (i.e. cost) does not provide a significant reward signal for returning to the distribution. This can be observed in the $H = 10$ plot where the performance of cost-only planning quickly drops as the discriminator is further refined on the expert data, decreasing the likelihood of randomly sampling into distribution. In this sense, the combination of cost and value operates as intended: *costing* is necessary for defining and staying inside the expert distribution, while *value* is necessary for generalizing the reward beyond the support of the expert elsewhere in the environment.

### D.5 TOWARDS HIGH-DIMENSIONAL TASKS FROM DEMONSTRATION WITH SAMPLED-BASED MPC

Figure 13 provides an experiment of MPAIL training an agent in the Isaac Lab implementation of the Ant-v2 environment as a step towards high-dimensional applications. As expected, MPPI's (vanilla) sampling procedure struggles to be competitive with policy-based optimization in higher-dimensional spaces. However, MPAIL demonstrates signs of life in enabling MPPI to optimize a state space otherwise considered extremely challenging for sample-based planning. Note that Isaac Lab's Ant implementation prescribes a 60-

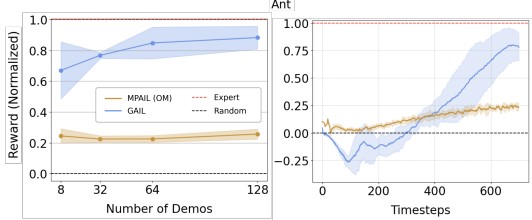

Figure 13: **Isaac Lab Ant-v2 Experiment with MPAIL (OM) and GAIL.**

dimensional observation and 8-dimensional action space, rather than Mujoco's 26-dimensional observation. Thus, the space is 120-dimensional for costing $c_\theta(s, s')$ and 80-dimensional for MPPI with a planning horizon of 10 timesteps. Despite this, a learned cost is capable of guiding a real-

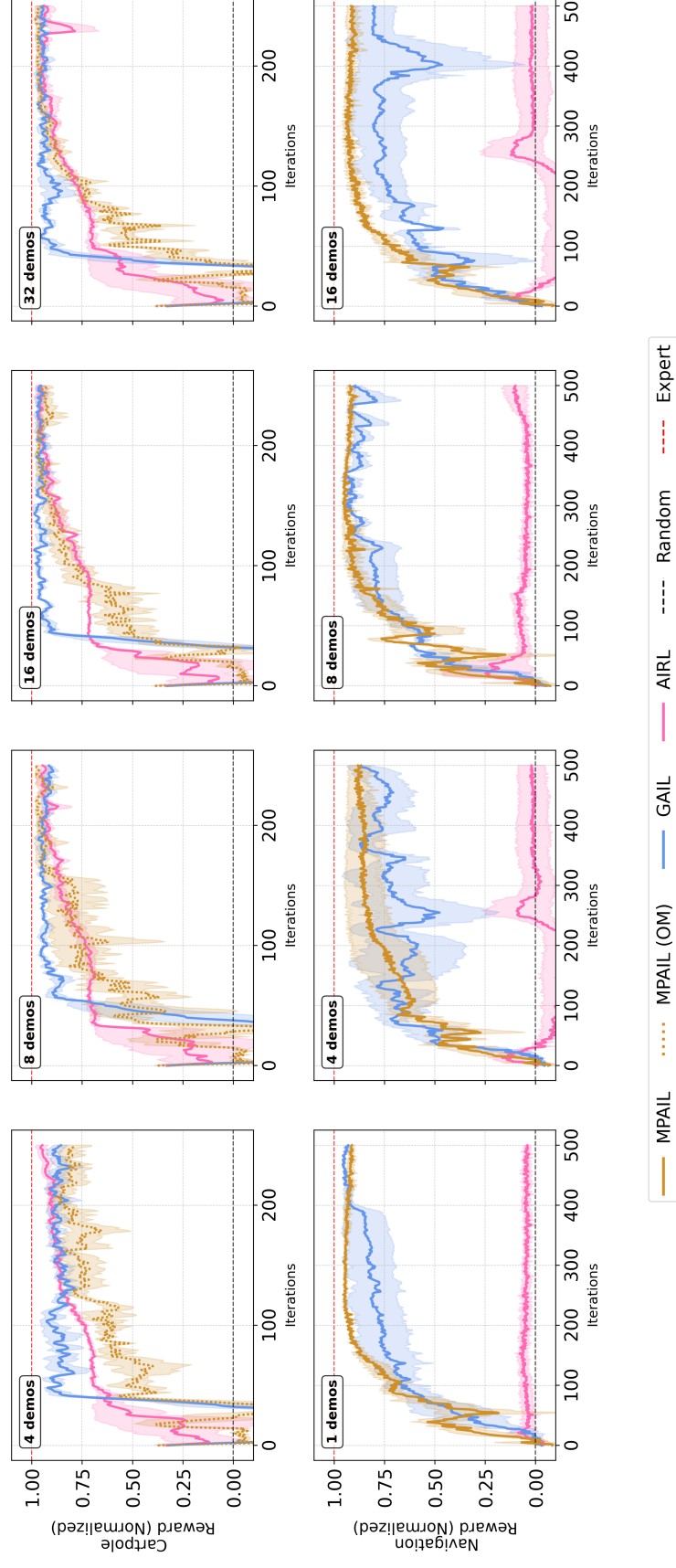

Figure 11: Benchmark results from Figure 7 de-aggregated across number of demonstrations. MPAIL results for the Cartpole task is differentiated **MPAIL (OM)** to clarify that its **M**odel is learned fully **O**nline.

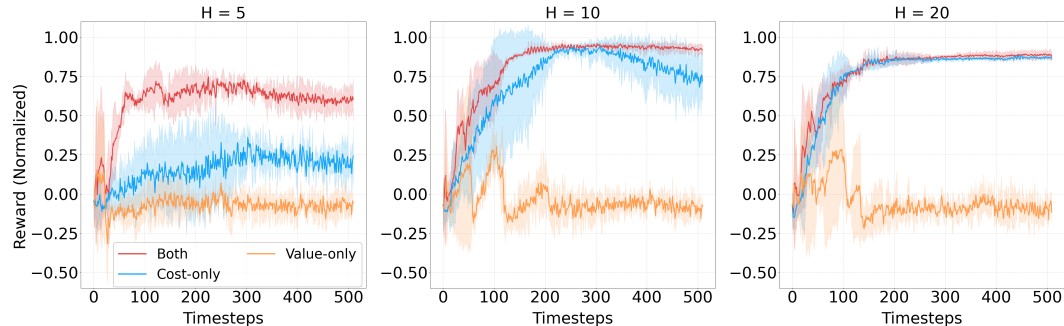

Figure 12: **Ablating single-step costs $c_\theta$ and value $V_\phi$ across different Horizon ($H$) lengths.** "Cost-only" experiments are performed by *not evaluating $V_\phi$* on the final state in each model rollout. "Value-only" experiments are performed by *not evaluating $c_\theta$* on the $H$-step state-transitions. See Algorithm 2, line 12 for exact usage.

time vanilla MPPI optimization to execute walking behaviors in the ant task, albeit slower, even from few demonstrations.

"Remembering" locally optimal policies through a learned policy-like proposal distribution may help planning capabilities generalize to higher-dimensional spaces. Additionally, modeling dynamics in latent-space and using model ensembles have been shown to significantly improve performance in model-based reinforcement learning (Hansen et al., 2022) and are promising directions for future work for high-dimensional tasks. Finally, Figure 10 illustrates a key takeaway of this framework: learning costs through MPAIL remains orthogonal to other works which seek to improve sample-based MPC through sampling (Xue et al., 2024; Vlahov et al., 2024; Sacks & Boots, 2023), optimization (Vlahov et al., 2024; Sacks et al., 2024), and dynamics (Hansen et al., 2024). We believe that integration of developments in MPC along with application-specific cost regularization (Finn et al., 2016) may be critical for exploring the full potential of planning from observation.

