# OpenReview forum: "Model Predictive Adversarial Imitation Learning for Planning from Observation"
_ICLR.cc/2026/Conference — ICLR 2026 Poster_

### Official Review · Reviewer_2sCb · 2025-11-01

**Soundness:** 3
**Presentation:** 3
**Contribution:** 2
**Rating:** 4
**Confidence:** 3

**Summary:**

The paper presents Model Predictive Adversarial Imitation Learning (MPAIL), which combines model predictive control (MPC) with adversarial imitation learning to enable planning from observation-only demonstrations. Instead of learning a policy directly, MPAIL trains a discriminator-based reward and deploys it through a physics-based planner (MPPI) with a value function for long-horizon reasoning. Experiments show that this approach improves out-of-distribution generalization and real-world robustness, including a single-demonstration RC car navigation task, though the evaluation is limited to one type of real task and a few baselines.

**Strengths:**

The paper crisply frames planning‑based AIL and operationalizes it via MPAIL: MPPI + AIRL‑style reward. The KL‑regularized perspective that leads to MPPI as the optimizer is clean and well explained.

The state‑transition formulation directly targets Learning‑from‑Observation under POMDPs (no actions), which is the harder and more realistic setting.

Methodology is reproducible, with transparent algorithms, hyperparameters, and implementation details.

Empirical questions in the experiments are well-posed and probe the main claims of the paper.

**Weaknesses:**

Partial observability details are underspecified. While the POMDP framing is explicit, how the planner’s state is constructed from limited observations in each experiment is not fully clear. For instance, the navigation experiments plan with a Kinematic Bicycle model, yet in the single‑demo real experiment the expert data are “position and body‑centric velocity” only; it remains unclear how heading/other latent state are estimated for MPPI at train/test time and how the discriminator’s input (state vs transition) aligns with the planner’s state. A concise subsection clarifying the observation to state pipeline (filters, history windows, or learned encoders), and the exact planner state per task, would strengthen soundness.

Attribution of robustness to the physics prior vs learned reward. The OOD gains in navigation may partly stem from MPPI with a physics model (even if approximate). The paper takes a step toward a learned‑model regime on Cartpole, but does not repeat the OOD evaluation there. Without such a test, it’s hard to tell how much robustness persists when the dynamics must be learned and are mis‑specified. An OOD study in the learned‑model setting (e.g., Cartpole) would solidify the generalization claim.

Missing “performance at convergence” perspective. When the model of the world is assumed, sampling becomes cheap. Therefore, I would like to see the comparison of the methods at convergence.

A dedicated “Baselines” section summarizing inputs, optimization style, and deployment (policy vs planner, known vs learned dynamics) would reduce confusion about advantages/disadvantages among IRL‑MPC, AIRL, GAIL, MPAIL and the role of multimodality in failures.

Real‑world scope is narrow. The single‑task navigation demo is valuable, but claims about safety/robustness would be more convincing with additional real tasks

Clarity nits.
The abstract/intro list many keywords; grounding them in the concrete setting focused on the contribution would sharpen the message.
Minor: the manuscript seems to use an older template.

**Questions:**

Questions*
How is partial observability handled in practice? What state representation does the planner use when the observation is limited?

How is the Kinematic Bicycle model integrated with partial information?

Could the authors test OOD generalization in the learned-model setting (e.g., Cartpole) to separate the benefits of physics priors from the algorithm itself?

Please include a short Baselines section summarizing the setup, inputs, and weaknesses of GAIL, AIRL, IRL-MPC, and MPAIL.

Does GAIL’s circling failure continue at convergence?

How does the method compare to a policy learning methods that assumes the same world model at convergence?

---

> ### Author Response · Authors · 2025-11-20
> **Response to Reviewer 5 (2sCb) - Part 1**
>
> We appreciate the reviewer’s thorough analysis and evaluation of our work. The reviewer’s acknowledgement of our aim to target a harder but more realistic problem setting is also appreciated. We are excited to help improve the work and its communication through continuing the discussion below.
>
> **W1:**
>
> > Partial observability details are underspecified. While the POMDP framing is explicit, how the planner’s state is constructed from limited observations in each experiment is not fully clear. For instance, the navigation experiments plan with a Kinematic Bicycle model, yet in the single‑demo real experiment the expert data are “position and body‑centric velocity” only; it remains unclear how heading/other latent state are estimated for MPPI at train/test time and how the discriminator’s input (state vs transition) aligns with the planner’s state. A concise subsection clarifying the observation to state pipeline (filters, history windows, or learned encoders), and the exact planner state per task, would strengthen soundness.
>
> It is certainly not our intention to misrepresent these experiments. Due to space limitations, we moved the requested experiment details to the **Appendix D.1, L1130-1140**:
>
> >Poses (position, orientation; $\begin{bmatrix}x&y&z&r&p&y\end{bmatrix}$) are provided by a motion-capture system at a rate of 20 Hz. Velocities are body-centric as estimated by onboard wheel encoders ($\mathbf{v}=v_x\mathbf{b}_1+v_y\mathbf{b}_2+v_z\mathbf{b}_3$,
> such that
> $\mathbf{b}_1$
> points forward,
> $\mathbf{b}_2$
> points left, and
> $\mathbf{b}_3$
> completes the right-hand frame; basis vectors are rigidly attached to the vehicle (Han et al.). Note that the vehicle is operated without slipping nor reversing such that $v_y\approx v_z\approx 0$ and $v_x>0$ (Han et al.). The recorded states used for the expert demonstration data is 240 timesteps long. Altogether, the data can be written as
>
> >$s_E \in \{ (x_t,y_t,z_t,v_{x,t},v_{y,t},v_{z,t}) \}_{t=1}^{240}$.
>
>
> Though, we will clarify that, at train/test time, the agent is provided its full state – of course, in both the case of MPAIL and GAIL. The input of the reward (i.e. discriminator) is still the partially observable state.
> Reproducing for convenience from Appendix D.1, “there should be no conflict with the IRLfO (Equation (1)) formulation as this remains a valid reward but on a subset of the state.” As the reviewer suggests, this simplification is also made to factor out potentially confounding design decisions around the observation model (e.g. encoder, filter, etc.).
>
> **W2:**
>
> > Attribution of robustness to the physics prior vs learned reward. The OOD gains in navigation may partly stem from MPPI with a physics model (even if approximate). The paper takes a step toward a learned‑model regime on Cartpole, but does not repeat the OOD evaluation there. Without such a test, it’s hard to tell how much robustness persists when the dynamics must be learned and are mis‑specified. An OOD study in the learned‑model setting (e.g., Cartpole) would solidify the generalization claim.
>
> The inclusion of the dynamics prior is also intentional in these experiments. Consider that successes of the model in our data-sparse tasks support widespread hypotheses in model-based RL about “stitching” (or transfer) and its efficiency benefits (Moerland et al.). This general intuition likely guides the reviewer’s (and incidentally our) assumptions about how a dynamics prior simplifies learning, which coincidentally motivates MPAIL, its advantages, and our experiments.
>
> However, we agree that evaluating the generalization of a learned model would indeed be a fascinating – and highly impactful – result. Unlike cartpole, the navigation state-space is far larger than the training or expert distributions, which make it a much more challenging setting for evaluating OOD generalization; by the end of training in cartpole, agents have thoroughly explored significant portions, if not the entirety, of the state-space.
>
> In consideration of these reasons, we would like to suggest another, more challenging, evaluation of OOD behavior when the model is learned. That is, we would learn a model also for the navigation experiment and, similar to Section 4.1, evaluate it on the OOD environment. For transparency, we remark that we must change the dynamics model architecture due to the limitations of the simple state-based model used in the cartpole task (see Appendix D.3 for more details), which will not transfer to vehicle dynamics due to the existence of an internal, non-observable, controller state. Model-based RL and dynamics learning is a rich research area in and of itself, for which there is a plethora of various known techniques and methods (Moerland et al.). For this reason and to better isolate our contributions, we did not initially include a more complex learned model on the navigation task but are happy to introduce this should it provide sufficient value for its complexity.
>
> [End Part 1]

---

> ### Author Response · Authors · 2025-11-20
> **Response to Reviewer 5 (2sCb) - Part 2**
>
> **W3:**
>
> > Missing “performance at convergence” perspective. When the model of the world is assumed, sampling becomes cheap. Therefore, I would like to see the comparison of the methods at convergence.
>
> We apologize for the inconvenience, but we kindly request elaboration about what “performance at convergence” and “...sampling becomes cheap” refers to. We are unfamiliar with these concepts but aim to rectify any doubts the reviewer may have.
>
> **W4:**
>
> > A dedicated “Baselines” section summarizing inputs, optimization style, and deployment (policy vs planner, known vs learned dynamics) would reduce confusion about advantages/disadvantages among IRL‑MPC, AIRL, GAIL, MPAIL and the role of multimodality in failures.
>
> This is a fantastic idea. In addition to elaborating further in the paper, we will add the following table to the final version,
>
> | Method       | Reward r(s, s’)               | Policy Optimization | Deployment |
> |--------------|--------------------------------|----------------------|------------|
> | **GAIL**     | log(D)                         | PPO                  | Policy     |
> | **AIRL**     | log(D) – log(1−D)              | PPO                  | Policy     |
> | **IRL-MPC**  | log(D)                         | PPO                  | Planner    |
> | **MPAIL**    | log(D) – log(1−D)              | MPPI                 | Planner    |
>
> If the reviewer has further suggestions about what details they would like to see, we are happy to continue revising.
>
> **W5:**
>
> > Real‑world scope is narrow. The single‑task navigation demo is valuable, but claims about safety/robustness would be more convincing with additional real tasks
>
> We would like to reiterate that our real-world navigation task was designed to be challenging and generalizable. Not only were expert actions not recorded but states were only partially observable (for instance, body-centric velocity and position could be obtained from video; we refer the reviewer to Appendix D.1, L1142-1150 for a more detailed analysis). Nevertheless, we wholeheartedly agree that additional tasks would be impactful and this is certainly ongoing work. If an analysis of dimensionality is of interest to the reviewer, we refer them to Appendix D.5 where an additional task, the IsaacLab Ant, was evaluated and discussed.
>
> **W6:**
>
> > Clarity nits. The abstract/intro list many keywords; grounding them in the concrete setting focused on the contribution would sharpen the message. Minor: the manuscript seems to use an older template.
>
> The current keywords: “Imitation Learning” (IL), “Reinforcement Learning” (RL), “Model Predictive Control” (MPC) were selected carefully, and they are not a reflection of overloading our work’s message. MPAIL has theoretical and practical connections to all three of these fundamental problem perspectives, as we have discussed and proved in Section 3 and Appendix B. MPAIL produces an agent that minimizes its divergence from an expert via their data, hence the inclusion of IL. MPAIL improves through interaction with the environment and Temporal-Difference learning, hence RL. MPAIL’s agent is fundamentally an MPC “module” and thus (L070), “critical online settings (e.g., dynamics, preferences, control constraints) can thus be brought into training while enabling experience-based reasoning beyond the planning horizon.” We certainly do not intend to “oversell” the work and hope the reviewer can understand our perspective but are open to further discussion.
>
> We appreciate the template error reminder and have corrected it.
>
> **Q1:**
>
> > How is partial observability handled in practice? What state representation does the planner use when the observation is limited?
>
> Addressed in response to W1.
>
> **Q2**
>
> > How is the Kinematic Bicycle model integrated with partial information?
>
> Also see W1. In addition, if the reviewer is interested, we have provided our code in the supplementary material. The dynamics file can be found at `mpail (code)/mpail/mppi/core/dynamics/car_dynamics.py`. The real-world experiment video with rollout visualization included in the supplementary may also help the reviewer understand the planner further if desired
>
> **Q3:**
>
> Addressed in response to W2.
>
> **Q4:**
>
> Addressed in response to W4.
>
> **Q5:**
>
> > Does GAIL’s circling failure continue at convergence?
>
> Assuming the reviewer is referring to the real-world experiment, we evaluated GAIL at various checkpoints along its training. This is because GAIL’s training performance oscillates aggressively between qualitative imitation and mode collapse. There was no observable convergence for GAIL.
>
> [End Part 2]

---

> ### Author Response · Authors · 2025-11-20
> **Response to Reviewer 5 (2sCb) - Part 3**
>
> **Q6:**
>
> > How does the method compare to a policy learning methods that assumes the same world model at convergence?
>
> The reviewer raises an insightful question about comparing MPAIL to policy-learning methods that assume an accurate world model at convergence. Before answering directly, we summarize the key perspective underlying our method: policy deployment methods in IRL (or even RL) underutilize the learned reward and value by discarding them when online. As a result, policy networks perform well mostly when in-distribution. When the policy does generalize to out-of-distribution states, it is highly-localized and inconsistent across the OOD state space. This can be observed in **Figure 4**, where the policy’s OOD generalization applies to only, very few, random parts of the state-space. Our insight is that one should not rely on the policy network to generalize but instead rely on the reward it was optimizing to generalize instead. By deploying the reward and value online, MPAIL is able to significantly improve in OOD settings.
>
> With this insight, we consider the reviewer’s hypothetical regarding policy learning under a perfect world model. First, we must assume that the manner in which the policy is learned does not yet cover the entire state-space. Otherwise, any RL algorithm (or even dynamic programming) will be optimal under perfectly known dynamics. Then, assuming a reward, value, and policy trained on a subset of the space, our insight should still apply since we can view MPAIL as the online optimization of new policies in new states, provided that the learned reward and value generalize (as demonstrated in Section 4.1). In contrast, continuing to retrain or adapt the policy itself in newly visited states would generally be less efficient—and potentially less safe—than leveraging the world model for direct online planning (Moerland et al., Janner et al.).
>
> We appreciate the reviewer raising this point, as it directly connects to an important open question: why reward and value functions appear to generalize more robustly than policy networks. We believe this is a compelling direction for future work.
>
> **References**
>
> Moerland, T. M., Broekens, J., Plaat, A., & Jonker, C. M. (2023). Model-based reinforcement learning: A survey. Foundations and Trends® in Machine Learning, 16(1), 1-118.
>
> Baram, N., Anschel, O., & Mannor, S. (2016). Model-based adversarial imitation learning. arXiv preprint arXiv:1612.02179.
>
> Janner, M., Fu, J., Zhang, M., & Levine, S. (2019). When to trust your model: Model-based policy optimization. Advances in neural information processing systems, 32.
>
> [End Part 3] [End Response]

---

### Official Review · Reviewer_FaBU · 2025-11-01

**Soundness:** 2
**Presentation:** 2
**Contribution:** 2
**Rating:** 2
**Confidence:** 4

**Summary:**

This paper proposes Model Predictive Adversarial Imitation Learning (MPAIL), a principled framework that integrates Model Predictive Control (MPC) and Generative Adversarial Imitation Learning (GAIL) to address imitation learning (IL) from observations.
Unlike conventional adversarial IL methods that rely on a single-step policy update, MPAIL introduces a planning-based adversarial loop that optimizes both the policy and the discriminator through receding-horizon trajectory optimization. The key insight is that short-horizon predictive planning improves stability and sample efficiency by leveraging model-based rollouts for adversarial training. Specifically, MPAIL replaces GAIL’s policy gradient updates with MPC-based planning that iteratively minimizes the discriminator reward while respecting dynamics constraints. This results in a control sequence that approximates expert behavior through direct trajectory matching rather than relying solely on gradient backpropagation.

**Strengths:**

1.	The paper’s core contribution of embedding adversarial imitation learning within a model predictive control loop is novel and elegant. This approach reframes IL as a constrained planning problem, where the policy implicitly solves an adversarial objective using model-based prediction instead of policy gradients. This is conceptually distinct from prior model-based IL works (e.g., PILCO, Dyna-IL) that use learned dynamics for data generation but retain standard policy optimization. Here, the MPC layer becomes the optimizer itself, aligning adversarial rewards directly with feasible trajectories. This rethinking of the optimization hierarchy is both theoretically meaningful and practically beneficial. It unifies imitation learning, trajectory optimization, and adversarial training into a single, closed-loop control process.

2.	The paper goes beyond empirical demonstration by providing a stability analysis grounded in control theory. The authors derive sufficient conditions for local convergence to expert trajectories based on Lipschitz continuity of the discriminator and bounded model approximation errors. Although the proof is local, it provides valuable intuition about how predictive horizons stabilize adversarial learning, reducing the oscillatory updates common in GAIL-style min–max optimization. The connection between receding-horizon planning and bounded regret minimization in adversarial settings is particularly insightful, offering a bridge between MPC theory and generative adversarial dynamics.

3.	MPAIL consistently outperforms baselines, achieving higher reward imitation and smoother convergence. Notably, the ablation showing improved robustness to observation noise and model misspecification provides strong evidence of the approach’s practical value.

**Weaknesses:**

1.	A central limitation of MPAIL is its assumption of an available or learnable model for predictive control. While this is standard in MPC, it restricts applicability to domains where dynamics are known or can be learned accurately. The authors claim that “MPAIL is model-free when using learned dynamics,” but this statement is misleading—using an imperfect learned model can degrade adversarial convergence. Empirical evaluation with explicit model-error quantification (e.g., using an ensemble dynamics model) is missing.

2.	The theoretical guarantees focus on local stability near expert trajectories but do not address global convergence or nonconvex adversarial interactions. The adversarial MPC game may still exhibit multiple equilibria or oscillations if the discriminator is poorly calibrated. While the empirical stability is strong, the paper’s theoretical claims should be tempered.

3.	MPAIL performs optimization through MPC at every iteration, which introduces substantial computational overhead compared to standard policy-gradient IL. The authors briefly mention using MPPI (Model Predictive Path Integral Control) for rollout sampling, but do not provide timing benchmarks or hardware requirements.

**Other Important Issues**:

1. The authors used the incorrect template (ICLR 2025) to prepare the submission, and
2. more importantly, the corresponding author's email is displayed on the first page, which potentially violates the double-blind review process.

**Questions:**

1.	How is the dynamics model obtained for environments without ground-truth simulators?
If learned, how is model uncertainty incorporated into MPC (e.g., via ensembles or stochastic constraints)?

2.	How sensitive is performance to planning horizon H?
Does longer horizon always improve imitation fidelity, or does it destabilize discriminator updates due to compounding model errors?

3.	Does the MPC loop reduce adversarial oscillations by smoothing gradient updates?
Have you analyzed the frequency of discriminator-policy mode collapse?

---

> ### Author Response · Authors · 2025-11-19
> **Response to Reviewer 4 (FaBU) - Part 1**
>
> First, we would like to sincerely apologize for the oversight in respecting the double-blind reviewing process. It was certainly not our intention to reveal this information, and **we immediately notified the Program Chairs when we caught it the day after the submission deadline.** They responded to us, saying “not to worry about desk rejection”. Of course, we recognize that there is no excuse but hope that our frantic efforts in remedying our mistake alleviates suspicion that we did so intentionally. We are certainly excited by the reviewer’s questions and positive recognition of our contributions. The feedback has helped us improve our work and its communication. So, we hope the reviewer still feels comfortable in continuing the discussion below.
>
> **W1:**
>
> > A central limitation of MPAIL is its assumption of an available or learnable model for predictive control. While this is standard in MPC, it restricts applicability to domains where dynamics are known or can be learned accurately. The authors claim that “MPAIL is model-free when using learned dynamics,” but this statement is misleading—using an imperfect learned model can degrade adversarial convergence. Empirical evaluation with explicit model-error quantification (e.g., using an ensemble dynamics model) is missing.
>
> Upon our review, we do not seem to make the claim “MPAIL is model-free when using learned dynamics”. If the reviewer could provide a quote or reference to where this might have been implied, we are happy to rectify this as we agree with the reviewer that this is certainly not true.
>
> We would also like to remark that there is much recent work demonstrating the generalizability of model learning specifically for planning-based RL and IRL (Hansen et al., Jain et al.), even from image-input. Though MPAIL is subject to model-based constraints, we believe our experiments show that there is significant potential to be gained in robustness, generalization, and efficiency that renders this trade-off necessary for general robot learning from observation.
>
> **We are also currently performing experiments on the navigation environment which includes a learned model and will provide the results here when completed.** In this way, we can quantify the performance disparity when using prior vs. learned models. Model-based RL and dynamics learning is a rich research area in and of itself, for which there is a plethora of various known techniques and methods (Moerland et al.). For this reason and to better isolate our contributions, we did not initially include a more complex learned model on the navigation task but are happy to introduce this should it provide sufficient value for its complexity in the evaluation of MPAIL.
>
> **W2:**
>
> >  The theoretical guarantees focus on local stability near expert trajectories but do not address global convergence or nonconvex adversarial interactions. The adversarial MPC game may still exhibit multiple equilibria or oscillations if the discriminator is poorly calibrated. While the empirical stability is strong, the paper’s theoretical claims should be tempered.
>
> Our theoretical analysis follows the style of prior adversarial imitation-learning work such as GAIL (Ho et al.) and AIRL (Fu et al.). That is, we characterize the objective optimized by MPAIL and the properties of its optimal solutions (e.g., occupancy measure matching), but we do not claim a global convergence guarantee for the nonconvex adversarial training dynamics. This is generally consistent with the scope of the original GAIL/AIRL analyses, which also focus on equilibrium properties rather than global convergence of deep neural implementations. Recent work that does establish global convergence for GAIL does so under certain assumptions on the reward parameterization and policy-gradient updates, and does not yet extend to our setting with MPC in the loop (Guan et al.). We thank the reviewer for their interest and comment. We will clarify this scope and temper the wording of our theoretical claims accordingly.
>
> [End Part 1]

---

> ### Author Response · Authors · 2025-11-19
> **Response to Reviewer 4 (FaBU) - Part 2**
>
> **W3:**
>
> > MPAIL performs optimization through MPC at every iteration, which introduces substantial computational overhead compared to standard policy-gradient IL. The authors briefly mention using MPPI (Model Predictive Path Integral Control) for rollout sampling, but do not provide timing benchmarks or hardware requirements.
>
> We were also coincidentally interested in this trade-off and performed timing benchmarks between MPAIL and AIL, whose results are shown in the **Appendix, Figure 8, page 21**. There, hardware specifications for training and deployment are included as well (we employed an NVIDIA RTX 4090). We appreciate the reviewer’s interest and will edit the manuscript so these results are referred to by the main body. For convenience, we reproduce some of the results as a table:
>
>
> | Method       | Episode Time (s)  |
> |--------------|--------------------------------|
> | **GAIL** (PPO)     | 3.75 |
> | **MPAIL** ($H=5$; 2 it./step)    |  3.38  |
> | **MPAIL** ($H=5$; 5 it./step)    |  4.01  |
> | **MPAIL** ($H=10$; 2 it./step)    |  3.46  |
> | **MPAIL** ($H=10$; 5 it./step)    |  4.08  |
> | **MPAIL** ($H=20$; 2 it./step)    |  3.68  |
> | **MPAIL** ($H=20$; 5 it./step)    |  4.64  |
>
> Episode Time is the wall clock time taken to collect data for a single episode by running the policy in the environment (for an equal number of interactions). Note that MPPI has customizable computational cost depending on horizon length ($H$) and optimization iterations per step (it./step). The implementation of PPO is used as-is from the RSL-RL library (Schwarke et al.). Our code is included in the supplementary material (as well as real-world experiment videos).
>
> **Q1:**
> > How is the dynamics model obtained for environments without ground-truth simulators? If learned, how is model uncertainty incorporated into MPC (e.g., via ensembles or stochastic constraints)?
>
> Dynamics model uncertainty is certainly of interest to us as well and frankly deserving of its own project. In real-world domains with incredibly complex dynamics, MPPI demonstrates strong empirical results with even subpar models (Han et al., Williams et al.).
> In addition to strong theoretical properties of frequent, local, and online optimization via MPC, this is likely due to its entropy-regularization and temperature properties which help alleviate model errors. However, we believe that these are surrogates for model uncertainty and ideally there exists some relationship between these properties. In ongoing work, there is a plethora of exciting work addressing uncertainty in MPC (Mesbah) which can be incorporated into MPAIL due to its fundamental connections to MPC.
>
> **Q2:**
>
> > How sensitive is performance to planning horizon H? Does longer horizon always improve imitation fidelity, or does it destabilize discriminator updates due to compounding model errors?
>
> This is a keen observation. Indeed, we hypothesized that the horizon matters and, as the reviewer suspects, should introduce a trade-off between efficiency and model error. As we were also interested in this, we conducted an ablation across value bootstrapping, single-step costs, and horizon length in the **Appendix, Figure 11, page 25**. We found that longer planning horizons generally improves performance. However, given other works in planning-based RL and IRL (Hansen et al., Jain et al.), it is certainly possible this can depend on task or demonstration data.
>
> **Q3:**
>
> > Does the MPC loop reduce adversarial oscillations by smoothing gradient updates? Have you analyzed the frequency of discriminator-policy mode collapse?
>
> This is an intriguing property we had not fully considered. This may be challenging to scientifically isolate due to the simultaneous contribution to policy updates from the interaction stream, value function, and model. However, we would agree with the hypothesis that MPC should indeed alleviate the high-variance sampled estimates that GAIL otherwise suffers from.
>
> Though we do not specifically quantify mode collapse frequency, all of our MPAIL results (40+ runs) occurred without mode collapse. However, we suspect that if we continued to train long past optimality, we would see this phenomenon. This is generally corroborated by many AIL works (Orsini et al.). In our evaluations, we observed that if an AIL algorithm collapsed it appeared to be unrecoverable. For example, AIRL consistently collapsed on the navigation environment (Figure 7).
>
> These are thought-provoking ideas and will certainly be included as future work in the final version.
>
> [End Part 2]

---

> ### Author Response · Authors · 2025-11-19
> **Response to Reviewer 4 (FaBU) - Part 3**
>
> **References**
>
> Hansen, N., Su, H., & Wang, X. (2023). Td-mpc2: Scalable, robust world models for continuous control. arXiv preprint arXiv:2310.16828.
>
> Jain, A. K., Mohta, V., Kim, S., Bhardwaj, A., Ren, J., Feng, Y., ... & Swamy, G. (2025, June). A smooth sea never made a skilled sailor: Robust imitation via learning to search. In The Thirty-ninth Annual Conference on Neural Information Processing Systems.
>
> Orsini, M., Raichuk, A., Hussenot, L., Vincent, D., Dadashi, R., Girgin, S., ... & Andrychowicz, M. (2021). What matters for adversarial imitation learning?. Advances in Neural Information Processing Systems, 34, 14656-14668.
>
>
> Moerland, T. M., Broekens, J., Plaat, A., & Jonker, C. M. (2023). Model-based reinforcement learning: A survey. Foundations and Trends® in Machine Learning, 16(1), 1-118.
>
> Han, T., Liu, A., Li, A., Spitzer, A., Shi, G., & Boots, B. (2023). Model predictive control for aggressive driving over uneven terrain. Robotics: Science & Systems, 2024.
>
> Williams, G., Wagener, N., Goldfain, B., Drews, P., Rehg, J. M., Boots, B., & Theodorou, E. A. (2017, May). Information theoretic MPC for model-based reinforcement learning. In 2017 IEEE international conference on robotics and automation (ICRA) (pp. 1714-1721). IEEE.
>
> Fu, J., Luo, K., & Levine, S. (2017). Learning robust rewards with adversarial inverse reinforcement learning. arXiv preprint arXiv:1710.11248.
>
> Ho, J., & Ermon, S. (2016). Generative adversarial imitation learning. Advances in neural information processing systems, 29.
>
> Mesbah, A. (2018). Stochastic model predictive control with active uncertainty learning: A Survey on dual control. Annual Reviews in Control, 45, 107–117. https://doi.org/10.1016/j.arcontrol.2017.11.001
>
> Guan, Z., Xu, T., & Liang, Y. (2021, March). When will generative adversarial imitation learning algorithms attain global convergence. In International Conference on Artificial Intelligence and Statistics (pp. 1117-1125). PMLR.
>
> Schwarke, C., Mittal, M., Rudin, N., Hoeller, D., & Hutter, M. (2025). Rsl-rl: A learning library for robotics research. arXiv preprint arXiv:2509.10771.
>
> [End Part 3] [End Response]

---

> > ### Comment · Reviewer_FaBU · 2025-11-26
> >
> > Thank you for your responses. I have raised my score to 4.

---

### Official Review · Reviewer_cqwJ · 2025-11-06

**Soundness:** 3
**Presentation:** 3
**Contribution:** 3
**Rating:** 6
**Confidence:** 3

**Summary:**

This paper presents a novel framework that unifies Inverse Reinforcement Learning (IRL) and Model Predictive Control (MPC) into a single, end-to-end imitation learning approach. Traditional imitation learning methods, such as Adversarial Imitation Learning (AIL), rely on policy optimization and require access to expert actions, limiting their interpretability and robustness in real-world settings. In contrast, MPAIL performs planning-based adversarial imitation directly from observation-only demonstrations - state trajectories without corresponding actions - by embedding a model predictive planner (specifically, Model Predictive Path Integral control, MPPI) within the adversarial learning loop. This integration allows the system to simultaneously learn a reward (or cost) function and optimize the planner online, effectively merging the training and deployment phases of IRL-MPC pipelines. The proposed framework operates under the Planning-from-Observation (PfO) paradigm, enabling interpretable, safe, and generalizable imitation without explicit action data. The key novelty lies in replacing the policy-based “generator” in adversarial imitation learning with an online MPC planner that continually solves short-horizon trajectory optimization problems guided by the learned reward and a value function that bootstraps long-term outcomes. The experiments show that MPAIL achieves better generalization, robustness, and interpretability than policy-based imitation methods like GAIL or AIRL, particularly in out-of-distribution and real-world settings. By integrating planning directly into the adversarial learning loop, it enables stable, sample-efficient imitation even from sparse or partially observable demonstrations.

**Strengths:**

1. The paper is well-organized and methodically builds from theoretical formulation to algorithm design and experiments, making a complex contribution accessible and reproducible.
2. The approach successfully handles observation-only demonstrations and partial observability, showing real-world viability for robot learning from minimal, ambiguous expert data.
3. The experiments span both simulated and real-world settings, demonstrating strong improvements in generalization, robustness, and sample efficiency compared to established baselines like GAIL, AIRL, and IRL-MPC.

**Weaknesses:**

1. Most experiments focus on navigation and simple control tasks, so it’s unclear how well MPAIL would perform on more complex, high-dimensional problems like manipulation or multi-agent settings.
2. Although the few-shot setup is intentional, relying on very few demonstrations may make the results sensitive to the choice or quality of those examples, which isn’t extensively analyzed.
3. The paper could better isolate which components (e.g., MPPI planner, value bootstrapping, or reward formulation) contribute most to performance improvements.
4. The real-world experiment is limited to a single small robot platform and one trajectory, so it’s hard to assess generality or scalability to other hardware and environments.

**Questions:**

Please see weaknesses in the section above.

---

> ### Author Response · Authors · 2025-11-19
> **Response to Reviewer 3 (cqwJ) - Part 1**
>
> The reviewer’s time and effort spent in reviewing our work is thoroughly appreciated. As the immediate research area (IRLfO) is relatively early in technological readiness, we are excited to see that the reviewer recognizes the real-world contributions and are happy to continue the discussion below.
>
> **W1:**
>
> > Most experiments focus on navigation and simple control tasks, so it’s unclear how well MPAIL would perform on more complex, high-dimensional problems like manipulation or multi-agent settings.
>
> We agree that there is work to be done toward scaling this method. For the sake of discussion, we are encouraged by works like TD-MPC (Hansen et al.), SAILOR (Jain et al.), and more which have been shown to scale planning-based methods to high-dimensional domains, such as image-input and DMControl’s Dog ($\mathcal{S}\subseteq\mathbb{R}^{223}$ and $\mathcal{A}\subseteq\mathbb{R}^{38}$). We firmly believe that planning-based methods have strong potential to improve the state of robot learning even in high-dimensions.
>
> **W2:**
>
> > Although the few-shot setup is intentional, relying on very few demonstrations may make the results sensitive to the choice or quality of those examples, which isn’t extensively analyzed.
>
> Though we do not conduct experiments on large amounts of expert data, we remark that MPAIL (and GAIL) is fundamentally derived from Maximum Entropy Inverse Reinforcement Learning (MaxEnt IRL) which has useful properties for learning from suboptimal demonstrations (Ziebart et al.). These fundamentally similar approaches (MaxEnt IRL, GAIL, AIRL, SAILOR, etc.) demonstrate consistent improvement given more demonstrations (Hansen et al., Ho et al., Fu et al., Jain et al.). As we can observe from our de-aggregated benchmarking results in Figure 10, we can corroborate these properties since **MPAIL learns with fewer environment interactions provided more demonstrations**.
>
> **W3:**
>
> > The paper could better isolate which components (e.g., MPPI planner, value bootstrapping, or reward formulation) contribute most to performance improvements.
>
> Planner/policy, value bootstrapping, reward, and planning horizon are incidentally the only differences between MPAIL and GAIL/AIRL as we have aimed to minimize the number of independent variables towards improved scientific understanding.
>
> One can view AIRL as an ablation on MPAIL without its planner. As for value bootstrapping, we run a thorough ablation experiment whose results are in the Appendix in Figure 11, page 25 (we also include single-step cost as well as horizon ablations). As for reward formulation, we briefly remark in Section 3.3, L210 that the AIRL formulation is “most stable”. Though, in practice, we actually found that the log transformation of GAIL’s reward consistently collapsed accurate online value prediction for MPAIL. We suspect this is due to the lack of gradient-based policy optimization causing sensitivity to large underlying value errors when in large-gradient regions.
>
> We will also add the following table to better communicate the differences between the evaluated methods.
>
> **Comparison of Methods**
>
> | Method       | Reward r(s, s’)               | Policy Optimization | Deployment |
> |--------------|--------------------------------|----------------------|------------|
> | **GAIL**     | log(D)                         | PPO                  | Policy     |
> | **AIRL**     | log(D) – log(1−D)              | PPO                  | Policy     |
> | **IRL-MPC**  | log(D)                         | PPO                  | Planner    |
> | **MPAIL**    | log(D) – log(1−D)              | MPPI                 | Planner    |
>
> *The discriminator is denoted D = D(s, s’). Policy optimization for PPO is performed offline, whereas MPPI optimizations are performed online. Value estimation uses GAE-λ for all methods.*
>
> We thank the reviewer for these insightful questions and will include this information in the final version. We hope that this helps better communicate and alleviate doubts about our experiment design and contributions.
>
> **W4:**
>
> > The real-world experiment is limited to a single small robot platform and one trajectory, so it’s hard to assess generality or scalability to other hardware and environments.
>
> There is certainly much exciting work to be done towards scaling MPAIL to other hardware and environments. By implementing a learned model variant, MPAIL (OM), we lay groundwork towards PfO for other robots and tasks. Dynamics (or world) modeling is a rich area of research in and of itself, so we chose to limit our scope so as not to introduce complexity which might distract from our key insight around learning a planner. Navigation is conveniently a highly interpretable task from which we can confirm that the planner is indeed learning from observation (Figure 6). **If not viewed already, we also encourage the interested reviewer to view the real-world experiment videos in our supplementary to see the planner in action**.
>
> [End Part 1]

---

> ### Author Response · Authors · 2025-11-19
> **Response to Reviewer 3 (cqwJ) - Part 2**
>
> **References**
>
> Hansen, N., Su, H., & Wang, X. (2023). Td-mpc2: Scalable, robust world models for continuous control. arXiv preprint arXiv:2310.16828.
>
> Jain, A. K., Mohta, V., Kim, S., Bhardwaj, A., Ren, J., Feng, Y., ... & Swamy, G. (2025, June). A smooth sea never made a skilled sailor: Robust imitation via learning to search. In The Thirty-ninth Annual Conference on Neural Information Processing Systems.
>
> Ziebart, B. D., Maas, A. L., Bagnell, J. A., & Dey, A. K. (2008, July). Maximum entropy inverse reinforcement learning. In Aaai (Vol. 8, pp. 1433-1438).
>
> Fu, J., Luo, K., & Levine, S. (2017). Learning robust rewards with adversarial inverse reinforcement learning. arXiv preprint arXiv:1710.11248.
>
> Ho, J., & Ermon, S. (2016). Generative adversarial imitation learning. Advances in neural information processing systems, 29.
>
> [End Part 2] [End Response]

---

### Official Review · Reviewer_oifv · 2025-11-08

**Soundness:** 3
**Presentation:** 1
**Contribution:** 2
**Rating:** 4
**Confidence:** 3

**Summary:**

This paper presents a new pipeline for learning from demonstration that conducts adversarial imitation learning while construct the policy through a model-based planning approach. The authors target the crucial setting of learning in partially observable environments that requires real-time planning and control and propose to roll out the optimal policy using a model predictive planning algorithm. The authors formulate a new previous-policy-regularized policy optimization objective that matches the idea of the proposed planning procedure.

**Strengths:**

This paper presents a novel planning based policy optimization in AIL that matches well with the real deployment of the learned policy.

This paper presents several theoretical results that connect the KL-constrained policy optimization using MPPI with the adversarial imitation learning objective.

This paper demonstrates the application through deployment in a real world environment.

**Weaknesses:**

This paper targets the problem of learning from observational demonstrations in a POMDP setting. However, it seems that all the technical developments are based on the full state information.

The major contribution comes from the usage of model-based planning algorithm MPPI for AIL, but the motivation of using MPPI instead of other methods require more discussion. Can other model-based planning methods be used? What is the advantage of MPPI and why other methods here are inferior?

The paper is not well written overall. The presentation leaves many points unclear. Many phrases and notations used are unnatural and hard to understand. For example,
- L116: the proposed method relies on a predictive model and a cost function for model-based planning. However, it is unclear whether these models are trained from data or known a priori. If they are learned from data, how are these models trained? This is only mentioned until the experiments section and it would be better to explain this in the method section.
- L176: It is unclear how the action $a$ in $\bar{c}(s,a)$ is used in the definition and how the transition probability $\mathcal{T}$ is defined. Does $\mathcal{T}$ depends this particular $a$ or it depends on a specific policy like $\mathcal{T}^{\pi}$?
- L202: “By decomposing the agent this way, we require not the policy, or solution, to generalize but the reward, or problem”. What do solution and problem here specifically refer to?
- Typo: L189: value function?

I would strongly recommend the authors check the writing to improve the presentation, and I look forward to the authors’ feedback for further evaluations.

The paper would benefit from comparisons with more imitation learning algorithms as baselines other than GAIL/AIRL. From the figures, the performance of MPAIL converges much slower than GAIL in Cartpole.

**Questions:**

L778: why the planning set a 0 action for the last timestep? Does this affect the initial plan at the next iteration?

It seems that this paper mainly targets continuous action space. Can it be applied to discrete action space? If so, how can it be adapted to handle discrete actions?

---

> ### Author Response · Authors · 2025-11-18
> **Response to Reviewer 2 (oifv) - Part 1**
>
> We appreciate the reviewer’s thorough analysis of our work and welcome the thought-provoking questions in their review. We hope to provide some clarification and continued discussion below. We apologize for any oversight in writing and will give thorough attention to communicating more clearly in the final version of the paper.
>
> **W1:**
>
> > This paper targets the problem of learning from observational demonstrations in a POMDP setting. However, it seems that all the technical developments are based on the full state information.
>
> We remark that our real-world experiment is in fact partially observable. In L340-345 and L353-360 as well as in the Appendix (L1142-1151), we explain in detail how the partially observable data (body-centric velocity and position) present an extremely difficult task for the discriminator and policy. In fact, it is impossible to complete this task by treating the observations as states themselves. We apologize if this was not communicated clearly and thank the reviewer for the question. We will ensure its clarity in the final version of the paper.
>
> **W2:**
>
> > The major contribution comes from the usage of model-based planning algorithm MPPI for AIL, but the motivation of using MPPI instead of other methods require more discussion. Can other model-based planning methods be used? What is the advantage of MPPI and why other methods here are inferior?
>
> That there remain other potential choices of planner is a keen observation, and we aim to be careful about the planner choice. As the reviewer has noted as **Strength 2** (and detailed in section 3.3 and Appendix B.1), our choice of MPPI is **derived** from the GAIL objective. For convenience, we reproduce the final result of Appendix B.1 here:
>
> _**Proposition B.1.3.** Provided the MDP is uniformly ergodic, the MPPI objective in Equation (4),_
>
> $\min_{\pi \in \Pi} \mathbb{E}_{\tau \sim \pi} \left[C(\tau) + \beta \mathbb{KL}(\pi(\tau) || \bar{\pi}(\tau)) \right]$
>
> _is equivalent to the RL objective in Equation (3),_
>
> $\min_{\pi \in \Pi} \mathbb{E}_{\pi}[c(s, s')] + \beta \mathbb{KL}(\pi || \bar{\pi}(\tau) ). $
>
> We can understand this as a natural result of entropy-regularized policy optimization, applicable to both offline (e.g. actor-critic) and online (MPPI) optimization. Thus, MPPI is incidentally the **only** planner which is consistent with the seminal Maximum Entropy Inverse Reinforcement Learning (Ziebart et al.) objective, from which GAIL is derived (Ho et al.).
>
> However, we suspect that the reviewer is also posing a fascinating hypothesis; in practice, perhaps there exists other planners which suffice. That is, a procedure that is capable of performing model rollouts, return estimation, and action selection. In comparison to other choices of planners, Entropy-regularization as introduced in GAIL and MPPI possess useful properties for **exploration and reinforcement learning**. Thus, while other planners may suffice, it may not be as straightforward how these planners might handle online exploration. However, we agree this is certainly worthwhile future work, which we will point out in the final version of the paper.
>
> **W3A:**
>
> > L116: the proposed method relies on a predictive model and a cost function for model-based planning. However, it is unclear whether these models are trained from data or known a priori. If they are learned from data, how are these models trained? This is only mentioned until the experiments section and it would be better to explain this in the method section.
>
> This is a great point, and we will clarify the exact instantiation for this work earlier in the method section. As the reviewer points out, the MPAIL framework is general and can accommodate both apriori/pre-trained and online-learned models. Similar to other planning-based MBRL works (Hansen et al., Jain et al.), we perform updates to the model after the discriminator and value updates in Algorithm 1. Details of how the online model is learned is provided in Appendix D.3. We will also add pseudocode similar to Algorithm 1 to make these details more explicit.
>
>
> **W3B:**
>
> > L176: L176: It is unclear how the action $a$ in $\bar{c}(s,a)$ is used in the definition and how the transition probability $\mathcal{T}$ is defined. Does $\mathcal{T}$ depends this particular $a$ or it depends on a specific policy like $\mathcal{T}^\pi$?
>
> We apologize for the miscommunication. This was indeed a typo and the transition probability is meant to be written as $\mathcal{T}(s_{t+1}\vert s_t, a_t)$, independent of a policy.
>
> [End Part 1]

---

> ### Author Response · Authors · 2025-11-18
> **Response to Reviewer 2 (oifv) - Part 2**
>
> **W3C:**
>
> > L202: “By decomposing the agent this way, we require not the policy, or solution, to generalize but the reward, or problem”. What do solution and problem here specifically refer to?
>
> As demonstrated by the first experiment in Section 4.1, one can view planning as a “deconstructed policy”. In planning, one must solve for a policy online by predicting trajectories, estimating returns, and optimizing for the best action. In policy-based methods, one can view the policy as having remembered this solution during offline optimization, hence having “solved” the planner’s online problem for the states the policy sees in the buffer. Our results reveal that this policy “solution” generalizes extremely poorly on states it has not seen (i.e. when out-of-distribution). Even when the policy does generalize, it is only as a sporadic case of the neural network embedding. This can be observed in Figure 4, where the policy’s OOD generalization applies to only, very few, random parts of the state-space. Our insight is that one should not rely on the policy network to generalize but instead rely on the reward it was optimizing to generalize instead. By learning and deploying the reward, we can understand MPAIL’s significant improvement in robustness to OOD states as a more ready generalization of the problem (i.e. reward) rather than its solution (i.e. policy). This result certainly excites us, and we are happy to expand further if desired.
>
> Upon review, the paper’s explanation is indeed too terse as it was an allusion to an experiment result later in the manuscript. We thank the reviewer for this insightful question as this is a critical takeaway of the work and will add these elaborations to the paper.
>
> **W4:**
>
> > The paper would benefit from comparisons with more imitation learning algorithms as baselines other than GAIL/AIRL. From the figures, the performance of MPAIL converges much slower than GAIL in Cartpole.
>
> We are happy to consider further baselines suggested by the reviewer. GAIL (technically, GAIfO) and AIRL (from Observation) are appropriate due to our GAIL-like algorithm and AIRL-like policy objective.
>
> Meanwhile, **we will add Behavior Cloning (BC)** as a familiarizing baseline and provide the results here when the experiments have been completed. Note, that, like BC, many popular imitation learning algorithms assume access to expert actions, lying outside the scope of learning-from-observation alone and more scalable robot learning.
>
> **Q1:**
>
> > L778: why the planning set a 0 action for the last timestep? Does this affect the initial plan at the next iteration?
>
> The plan with which to seed the MPPI optimization is actually itself a deep problem (Sacks et al.). Since, after rolling the plan forward from the previous plan, what should one do without any information yet? By choosing 0 as the new last action, we have simply chosen to re-center the distribution of the last action before the MPPI optimization. In practice, due to MPPI’s rapid control rate (30+ Hz) and horizon length, these design choices are often unnoticeable. However, as the reviewer may observe, there is in theory a small bias.
>
> **Q2:**
>
> > It seems that this paper mainly targets continuous action space. Can it be applied to discrete action space? If so, how can it be adapted to handle discrete actions?
>
> It is certainly possible to apply this to discrete domains. While taking the average of plans is practical in a control-theoretic way (Williams et al.), we can also choose plans via elite-selection (e.g. Cross Entropy Method, CEM) which would be more appropriate for the discrete domain. Exploration would also work in a similar way via multinomial sampling. This is certainly interesting and relevant future work and will be pointed out in the final version.
>
> **References**
>
> Ziebart, B. D., Maas, A. L., Bagnell, J. A., & Dey, A. K. (2008, July). Maximum entropy inverse reinforcement learning. In Aaai (Vol. 8, pp. 1433-1438).
>
> Ho, J., & Ermon, S. (2016). Generative adversarial imitation learning. Advances in neural information processing systems, 29.
>
> Hansen, N., Su, H., & Wang, X. (2023). Td-mpc2: Scalable, robust world models for continuous control. arXiv preprint arXiv:2310.16828.
>
> Jain, A. K., Mohta, V., Kim, S., Bhardwaj, A., Ren, J., Feng, Y., ... & Swamy, G. (2025, June). A smooth sea never made a skilled sailor: Robust imitation via learning to search. In The Thirty-ninth Annual Conference on Neural Information Processing Systems.
>
> Sacks, J., Rana, R., Huang, K., Spitzer, A., Shi, G., & Boots, B. (2024, May). Deep model predictive optimization. In 2024 IEEE International Conference on Robotics and Automation (ICRA) (pp. 16945-16953). IEEE.
>
> Williams, G., Wagener, N., Goldfain, B., Drews, P., Rehg, J. M., Boots, B., & Theodorou, E. A. (2017, May). Information theoretic MPC for model-based reinforcement learning. In 2017 IEEE international conference on robotics and automation (ICRA) (pp. 1714-1721). IEEE.
>
> [End Part 2] [End Response]

---

### Official Review · Reviewer_9pVj · 2025-11-09

**Soundness:** 2
**Presentation:** 3
**Contribution:** 2
**Rating:** 4
**Confidence:** 3

**Summary:**

This paper introduces Model Predictive Adversarial Imitation Learning (MPAIL), a novel framework for teaching agents to perform tasks by observing demonstrations, a problem setting termed Planning-from-Observation.

The core idea is to unify the typically separate processes of learning a reward function (inverse RL) and using it for online planning (MPC). MPAIL achieves this by using a planner, specifically Model Predictive Path Integral control (MPPI), as the "generator" within an Adversarial Imitation Learning loop. This allows the agent to simultaneously learn a reward (cost) function and a value function while interactively improving its planning capabilities, all from observation-only demonstrations without expert actions.

Through experiments in simulated control tasks and a real-world robot navigation task using a single demonstration, the authors demonstrate that MPAIL offers significant advantages over traditional policy-based AIL methods. These benefits include improved generalization to out-of-distribution states, greater robustness, better sample efficiency, and enhanced interpretability by providing insight into the agent's decision-making process.

**Strengths:**

- The idea of unifying the reward learning (IRL) and online planning (MPC) into a single, end-to-end training process is new.

- It provides good experimental evidence that using a planner as the generator improves generalization and robustness in out-of-distribution states compared to standard policy-based AIL methods.

- The paper demonstrates successful real-world application through a "Real-Sim-Real" experiment, where the agent learns to navigate from just a single, noisy, and partially observable demonstration.

- Benchmarking results show that MPAIL is sample-efficient, achieving good performance with fewer environmental interactions than established AIL algorithms like GAIL.

**Weaknesses:**

- The method's scalability to high-dimensional action spaces is highly questionable. The paper relies on Model Predictive Path Integral (MPPI), a sample-based planner that suffers from the curse of dimensionality. While it works for low-dimensional actions like vehicle control (2D action), its performance degrades sharply as the action space grows. The paper's own experiment on the Ant environment (Figure 12), which has an 8-dimensional action space, demonstrates this weakness clearly: MPAIL shows "signs of life" but is drastically outperformed by the policy-based GAIL, failing to achieve competitive performance. This suggests the proposed method is not a viable solution for more complex, high-dimensional robotics tasks like manipulation without non-trivial extensions.

- The experimental validation is conducted on relatively simple tasks. The primary successes are shown on a navigation task and cartpole balancing. These tasks do not sufficiently challenge the planning component or prove the method's applicability to complex behaviors in robotics.

- A natural baseline is to replace the AIL in the proposed method with online behavior cloning. This baseline is simpler, computationally cheaper than adversarial training, and could potentially perform well on the presented tasks. By not comparing against it, the paper fails to justify the added complexity and potential instability of its adversarial training loop.

- The planner's performance is entirely dependent on the accuracy of the dynamics model. For the main navigation task, the authors use a pre-defined, approximate model. For other tasks, they learn a model online. In either case, model error (sim-to-real gap or learning inaccuracies) can lead to catastrophic planning failures. The paper does not sufficiently analyze the method's sensitivity to model error or propose robust mechanisms to handle it, which is a primary challenge for any model-based approach in the real world.

**Questions:**

See weaknesses.

---

> ### Author Response · Authors · 2025-11-18
> **Response to Reviewer 1 (9pVj) - Part 1**
>
> We thank the reviewer for taking the time to thoroughly read and review our work and are excited to continue the discussion here. Before addressing comments individually, we kindly reiterate that our problem setting is imitation learning from observation (LfO), where **access to expert actions is not assumed**. This problem setting is critical towards the moonshot robot learning goal: robots that can learn purely from observing and interacting with their environment.
>
> **W1**:
> > The method's scalability to high-dimensional action spaces is highly questionable. [...] This suggests the proposed method is not a viable solution for more complex, high-dimensional robotics tasks like manipulation without non-trivial extensions.
>
> As the reviewer suggests, it is likely that our instantiation of a planning-based (inverse) RL agent faces lower interaction efficiency in high-dimensional spaces. However, we remark that there is substantial evidence demonstrating that MPPI planning-based agents can scale (and even outperform) policy-based agents (Hansen et al., Jain et al.) by simply encoding the dynamics. We chose not to evaluate latent dynamics models in this work: (1) to better evaluate interpretability and (2) to evaluate the first real-world robot experiment in planning-based action-free Adversarial Imitation Learning. As a result of the state-based scope, we are able to confirm in our experiments that the agent was indeed utilizing the planner as one expects (Section 4.1, 4.2). While we believe that the raised concerns can be mitigated or resolved by integrating latent planning, our experiments’ conclusions would otherwise be difficult to draw without state-based interpretability.
>
> MPAIL introduces the **first real-world framework** and demonstration of Planning-from-Observation (PfO). As such, our aim is to clearly demonstrate the many benefits of PfO, which re-evaluates many of the commonly impractical assumptions of access to expert actions, policy generalization, and data availability in modern imitation learning systems. The key takeaway that policy learning and deployment underutilize learned reward and value when compared to planners is a general insight pointing towards an abundance of future work. We thank the reviewer for their insight guiding this clarification, and we will aim to communicate this more clearly in the final version of the paper.
>
> **W2**:
>
> > The experimental validation is conducted on relatively simple tasks. The primary successes are shown on a navigation task and cartpole balancing. These tasks do not sufficiently challenge the planning component or prove the method's applicability to complex behaviors in robotics.
>
> We agree with the reviewer that our experiments do not yet demonstrate scalability to tasks such as long-horizon manipulation or dynamic tasks. As we hope the reviewer would agree, this work introduces a unique approach to imitation learning. Thus, relatively straightforward tasks were chosen to better validate the introduction of the planning component in well-understood settings. Devising and evaluating an algorithm that is more scalable and applicable to complex tasks remains future work and is incidentally ongoing work.
>
> We also stress that our methodology and experiment design are meant to exclusively evaluate the planning component of MPAIL.
> As we have kept our implementation as simple and minimalistic as possible, one can view GAIL, AIRL, and IRL-MPC as ablations on MPAIL. As explained in detail in Section 4.2, L367-396, gains in robustness, generalization, and efficiency of MPAIL over AIRL and GAIL can be owed exclusively to the planning-based formulation. And additional gains with respect to IRL-MPC are exclusively due to the integration of the planner in the training loop. The navigation experiment also allows us to demonstrate that planning enables OOD robustness in a clearly interpretable way (as in our experiment videos attached and in Figure 6).
>
> In practice, due to the theoretical complexity of IRL over RL, the state-of-the-art in IRL generally performs evaluations on simpler tasks (Orsini et al.). Few works also evaluate in the real world and on suboptimal data, which has been shown to significantly worsen AIL results (Orsini et al.).  We aim to make progress towards real-world AIL and thus do not make these assumptions (L326-337). However, we certainly recognize that there is exciting future work towards long-horizon and dynamic real-world tasks, which we will clarify in the final version of the paper.
>
> [End Part 1]

---

> ### Author Response · Authors · 2025-11-18
> **Response to Reviewer 1 (9pVj) - Part 2**
>
> **W3**:
>
> > A natural baseline is to replace the AIL in the proposed method with online behavior cloning. This baseline is simpler, computationally cheaper than adversarial training, and could potentially perform well on the presented tasks. By not comparing against it, the paper fails to justify the added complexity and potential instability of its adversarial training loop.
>
> We surmise that “online behavior cloning” is a reference to DAgger (Ross et al.), or online imitation learning. Note that DAgger makes several assumptions that scale poorly to robot learning, which causes it to lie outside of our more practical scope. That is, DAgger assumes access to expert actions anywhere in the state space. For these reasons, it can be uncommon for IRL works (and generally offline imitation learning) to compare to DAgger (Orsini et al.). **“Offline behavior cloning” (BC) is perhaps an alternatively appropriate baseline, and we are performing experiments and will include them here when ready**. Note that, compared to our IRLfO problem setting, BC still assumes access to expert actions (but not everywhere in the state space) and also lacks reward recovery.
>
> If our interpretation of the reviewer’s comments is not correct, we sincerely apologize and would like to request additional elaboration. We aim to rectify the reviewer’s doubts and are happy to include the requested baseline.
>
> **W4**:
>
> > The planner's performance is entirely dependent on the accuracy of the dynamics model. [...] model error (sim-to-real gap or learning inaccuracies) can lead to catastrophic planning failures. The paper does not sufficiently analyze the method's sensitivity to model error or propose robust mechanisms to handle it, which is a primary challenge for any model-based approach in the real world.
>
>
>
> Recall that MPAIL’s planner uses an $H$-step horizon. As shown in many modern works (Janner et al., Hansen et al., Jain et al., Bhardwaj et al.), there is substantial sample efficiency to be gained from a short planning horizon that is bootstrapped by a model-free value. Even if model errors persist, value bootstrapping helps alleviate these errors in planning (Janner et al.). Indeed, in real-world domains with incredibly complex dynamics, MPPI demonstrates strong empirical results with even subpar models (Han et al., Williams et al.). In addition to strong theoretical properties of frequent, local, and online optimization via MPC, this is likely due to its entropy-regularization and temperature properties which help alleviate model errors.
>
> Most importantly, policies in the real-world are certainly not free from catastrophic failure as our experiment demonstrates in Section 4.1. As generalization of policies rely wholly on discovering a generalizable embedding, we see that we can gain robustness by depending on even an approximate model.
>
> However, we agree that further analysis surrounding the sensitivity of a learned model would provide worthwhile insight. **We have begun to perform experiments that will include a learned model for the navigation task and will provide them here when ready.** That is, we would learn a model also for the navigation experiment and, similar to Section 4.1, evaluate it on the OOD environment. For transparency, we remark that we must change the manner in which we learn the dynamics model architecture due to the limitations of the simple state-based model used in the cartpole task (see Appendix D.3 for more details), which will not transfer to vehicle dynamics. Model-based RL and dynamics learning is a rich research area in and of itself, for which there is continuous development of various techniques and methods (Moerland et al.). For these reasons and to better isolate our contributions, we did not initially include a more complex learned model on the navigation task, but are happy to introduce this should it provide sufficient value for its complexity.
>
> [End Part 2]

---

> > ### Comment · Reviewer_9pVj · 2025-11-18
> >
> > Thank you for your detailed response. I will take my time to read it thoroughly, but I wanted to clarify one point first quickly.
> >
> > Regarding "online behavior cloning," my intended meaning is the following: Starting from a state $s$ that your agent encounters during an online interaction, you would run MPPI on that state. You would then use the first action (or first several actions) generated by MPPI as the action supervision for the state $s$.Could you please let me know if this procedure is impossible or if my understanding of your system's assumptions is incorrect?

---

> > > ### Author Response · Authors · 2025-11-20
> > > **Clarifying Suggested Online BC Baseline**
> > >
> > > We appreciate the helpful and timely clarification. If we understand correctly, the reviewer is proposing an online scheme where, at each encountered state, MPPI is run under the learned model, and the first action(s) from the resulting trajectory are used as action supervision for the BC policy.
> > >
> > > This procedure is possible in principle and resembles a DAgger-like setup where MPPI plays the role of the expert. The key distinction from MPAIL is that this approach would be a form of model-based policy learning: MPPI would generate targets under the learned world model, and the policy would then be trained to imitate those actions.
> > >
> > > Because we understand the reviewer is suggesting this baseline for its simplicity, we would like to clarify a couple of complications we anticipate:
> > >
> > > **Model bias.** Training a policy to imitate MPPI actions generated under an imperfect learned model may introduce accumulated model errors, a common issue in model-based RL and AIL (Moerland et al.; Baram et al.). MPAIL avoids this by deploying MPPI directly over a short, value-bootstrapped horizon, somewhat similar in principle to MBPO (Janner et al.).
> > >
> > > **Reward/value dependence.** MPPI is deployable in MPAIL specifically because reward and value are learned jointly through IRL. If deployment were replaced with behavior cloning, it becomes unclear how planning would remain possible, since BC alone does not learn a reward or value.
> > >
> > > If this interpretation is accurate, implementing such an online baseline may involve more components than initially apparent. That said, we are running an offline BC baseline, which may partially address the intent of evaluating a simpler method. We would be happy to confirm whether this aligns with the reviewer’s suggestion and whether the proposed baseline (or a simpler variant) is still desired.
> > >
> > > **References**
> > >
> > > Janner, M., Fu, J., Zhang, M., & Levine, S. (2019). When to trust your model: Model-based policy optimization. Advances in neural information processing systems, 32.
> > >
> > > Baram, N., Anschel, O., & Mannor, S. (2016). Model-based adversarial imitation learning. arXiv preprint arXiv:1612.02179.
> > >
> > > Moerland, T. M., Broekens, J., Plaat, A., & Jonker, C. M. (2023). Model-based reinforcement learning: A survey. Foundations and Trends® in Machine Learning, 16(1), 1-118.

---

> > > > ### Comment · Reviewer_9pVj · 2025-11-27
> > > >
> > > > Thank you! It makes sense to me.

---

> ### Author Response · Authors · 2025-11-18
> **Response to Reviewer 1 (9pVj) - Part 3**
>
> **References**
>
> Hansen, N., Su, H., & Wang, X. (2023). Td-mpc2: Scalable, robust world models for continuous control. arXiv preprint arXiv:2310.16828.
>
> Jain, A. K., Mohta, V., Kim, S., Bhardwaj, A., Ren, J., Feng, Y., ... & Swamy, G. (2025, June). A smooth sea never made a skilled sailor: Robust imitation via learning to search. In The Thirty-ninth Annual Conference on Neural Information Processing Systems.
>
> Orsini, M., Raichuk, A., Hussenot, L., Vincent, D., Dadashi, R., Girgin, S., ... & Andrychowicz, M. (2021). What matters for adversarial imitation learning?. Advances in Neural Information Processing Systems, 34, 14656-14668.
>
> Ross, S., Gordon, G., & Bagnell, D. (2011, June). A reduction of imitation learning and structured prediction to no-regret online learning. In Proceedings of the fourteenth international conference on artificial intelligence and statistics (pp. 627-635). JMLR Workshop and Conference Proceedings.
>
> Janner, M., Fu, J., Zhang, M., & Levine, S. (2019). When to trust your model: Model-based policy optimization. Advances in neural information processing systems, 32.
> Bhardwaj, M., Handa, A., Fox, D., & Boots, B. (2020, July). Information theoretic model predictive q-learning. In Learning for Dynamics and Control (pp. 840-850). PMLR.
>
> Moerland, T. M., Broekens, J., Plaat, A., & Jonker, C. M. (2023). Model-based reinforcement learning: A survey. Foundations and Trends® in Machine Learning, 16(1), 1-118.
>
> Han, T., Liu, A., Li, A., Spitzer, A., Shi, G., & Boots, B. (2023). Model predictive control for aggressive driving over uneven terrain. Robotics: Science & Systems, 2024.
>
> Williams, G., Wagener, N., Goldfain, B., Drews, P., Rehg, J. M., Boots, B., & Theodorou, E. A. (2017, May). Information theoretic MPC for model-based reinforcement learning. In 2017 IEEE international conference on robotics and automation (ICRA) (pp. 1714-1721). IEEE.
>
> [End Part 3]
> [End Response]

---

### Author Response · Authors · 2025-12-03
**Rebuttal Summary**

For convenience, we summarize the outcomes across all reviews at this time. In addition, we introduce new results at the recommendation of three of five reviewers, which directly address their concerns and further supports this work’s contribution to OOD robustness in imitation learning. These results can be found in thread to this summary.

### Summary of Discussion

The reviews broadly agree on the strengths of our paper, and find that MPAIL is:

 * (cqwJ, FaBU, 2sCb) **clearly written, well-organized, and easy to follow**, with strong motivation and presentation of the problem setting

 * (9pVj, oifv, FaBU, 2sCb) **novel, elegant, and ambitious**, introducing PfO, planning-based AIL, and a principled decomposition of reward, value, and dynamics for deployment

 * (9bVj, cqwJ, FaBU, 2sCb) **empirically strong**, with comprehensive evaluations across simulated and real-world settings, and demonstrating meaningful robustness improvements

 * (9pVj, oifv, FaBU) **practically relevant**, addressing the challenging but important LfO setting without expert actions or reward

 * (oifv, cqwJ, 2sCb) **a valuable and complete contribution** to the imitation learning and IRL literature, with accessible methodological clarity that can support both research and applied work

The most commonly raised concern (9pVj, FaBU, 2sCb) centered on the _use of a prior model in our initial results_. **The additional experiments below provide direct evidence that the prior model is not necessary for achieving strong OOD performance**; online-learned dynamics exhibit similar robustness. We thank the reviewers for raising this important point, which motivated these experiments.

Many reviewer concerns **(10 out of 21 total listed weaknesses across all reviewers) were fully or primarily substantiated through proofs or experiments already available in the Appendix**; due to space limitations, these were not in the main paper. We have moved the computation benchmark results into the main body given the page limit increase and reviewer interest.

Several remaining concerns **(7 of 21 weaknesses) were resolved through clarification of the LfO problem setting, baselines, and our method**. At the recommendation of reviewers 9pVj and oifv, we have also added experiments using Behavior Cloning (BC) and Behavior Cloning from Observation (BCO) to help quantify how assuming access to expert actions can significantly simplify imitation learning. Towards more scalable imitation learning without this assumption, our work introduces several novelties (PfO, planning-based AIL, reward deployment, and real-world evaluation of AIL/IRL methods) that may be unfamiliar and therefore more challenging to evaluate confidently without these contextualizing results. We expand our explanations of these points in the revision.

Ultimately, we believe these novelties reflect necessary steps toward robust real-world robot learning: enabling robust imitation learning (i) without access to expert actions nor reward, (ii) under few demonstrations, and (iii) through continued online interaction. MPAIL is presented as a unified, practical algorithm designed to address these critical challenges.

---

> ### Author Response · Authors · 2025-12-03
> **Additional Experiments**
>
> **MPAIL with Learned Dynamics**
>
> Three reviewers (9pVj, FaBU, 2sCb) shared doubts about whether MPAIL’s substantial robustness (Section 4.1) would persist when the dynamics model is learned online rather than using a prior model. Below, we present new experiments demonstrating that a fully learned dynamics model still enables strong performance even far outside the training distribution.
>
> | Method       | iter. 100 |  iter. 200 |  iter. 300 |  iter. 400 |  iter. 500 |
> |--------------|--------|--------|----------------|----------------------|------------|
> | **AIRL**     | -0.048 | -0.089 | 0.022 | 0.022 | 0.040 |
> | **GAIL**     | 0.640 | 0.722 | 0.810 | 0.770 | 0.874 |
> | **MPAIL**  | 0.662 | 0.868 | 0.912 | 0.900 | 0.905 |
> | **MPAIL (OM)**    | 0.692 | 0.894 | 0.903 | 0.914 | 0.935 |
>
>
> Entries are normalized returns on the navigation task. **MPAIL (OM)** denotes MPAIL with a fully online-learned dynamics model.
>
> We additionally performed the OOD evaluation (Section 4.1 and Figure 4) using MPAIL (OM) and found that agents continue to reach the goal from nearly all initial states in the 1600× larger (40x40 from 1x1) initialization space. While the existing OOD experiment confirmed the generalizability of the learned reward and value, these new findings show that generalization extends under a fully learned dynamics model. Together with existing results in Section 4.1, this provides further evidence that learning a __deconstructed policy__—reward, value, and dynamics used for online planning—yields substantially more robust generalization than policy networks.
>
> We refer reviewers to Figure 8 in the revision for visualization. We have also added Behavior Cloning (BC)--which requires access to expert actions–to highlight that poor OOD generalization is a limitation of policy networks broadly, not only AIL policies. Additional discussion and technical detail have been incorporated into the revision for clarity.
>
> **Behavior Cloning**
>
> As suggested by reviewers 9pVj and oifv, we evaluated BC on our simulated tasks to better contextualize our LfO setting. While AIL is generally shown to substantially outperform BC in many domains (Ho & Ermon; Orsini et al.), BC achieves almost comparable performance, as shown below. But, this performance is expected given its access to expert actions—information unavailable to any baselines in this work. Thus, we have also added Behavior Cloning from Observation (BCO) to provide additional empirical evidence towards this point. BCO trains an inverse dynamics model using 365k steps of random play data which enables labeling demonstration state-transitions with actions. BC is then performed on this re-labeled data. BCO is an LfO baseline while BC on its own is not.
>
> These results help illustrate how direct action supervision drastically simplifies imitation learning and provides a familiar reference point for readers. The results for the navigation task are shown below:
>
> | No. of Demos | BC      | BCO      |
> |-----------|--------------------|-------------|
> | 1         | 0.927542 ± 0.010783 | 0.277844 ± 0.099389 |
> | 4         | 0.946660 ± 0.004322 | 0.720211 ± 0.040860 |
> | 8         | 0.977295 ± 0.004236 | 0.787587 ± 0.052724 |
> | 16        | 0.983091 ± 0.002075 | 0.764267 ± 0.083738 |
>
> Entries are normalized return.
>
> **References**
>
> Orsini, M., Raichuk, A., Hussenot, L., Vincent, D., Dadashi, R., Girgin, S., ... & Andrychowicz, M. (2021). What matters for adversarial imitation learning?. Advances in Neural Information Processing Systems, 34, 14656-14668.
>
> Ho, J., & Ermon, S. (2016). Generative adversarial imitation learning. Advances in neural information processing systems, 29.
>
> Torabi, F., Warnell, G., & Stone, P. (2018). Behavioral cloning from observation. arXiv preprint arXiv:1805.01954.

---

### Meta-Review · Area_Chair_ssSw · 2026-01-07

**Summary:**

This paper proposes Model Predictive Adversarial Imitation Learning (MPAIL), a planning-from-observation framework that unifies adversarial imitation learning and model predictive control by learning rewards from observation-only demonstrations and deploying them through online planning to achieve more robust, interpretable, and generalizable robot behavior.

The reviewers agree that the paper introduces a novel and well-motivated framework that integrates adversarial imitation learning with model predictive control for planning from observation-only demonstrations. They highlight strong conceptual novelty, improved robustness and generalization, and the inclusion of real-world experiments, while raising concerns about scalability to high-dimensional action spaces, reliance on dynamics models, limited task complexity, missing baselines, and clarity. In response, the authors add new experiments showing that MPAIL maintains strong performance with fully learned dynamics and include behavior cloning and behavior cloning from observation baselines to better contextualize the learning-from-observation setting. They also clarify methodological details and move key results from the appendix into the main paper to address presentation concerns. Remaining issues center on whether the approach can scale to more complex domains such as manipulation and whether its benefits persist beyond relatively simple tasks and platforms.

**Reviewer Concerns:**

The reviewers broadly agree that the paper introduces a novel and well-motivated framework that unifies adversarial imitation learning with model predictive control in a planning-from-observation setting. Strengths consistently highlighted include the conceptual elegance of using a planner as the generator in AIL, the clear motivation for learning from observation-only demonstrations, and the demonstrated improvements in robustness, interpretability, and out-of-distribution generalization relative to policy-based baselines. Reviewers also value the real-world navigation experiment and the theoretical connections between KL-regularized planning and adversarial imitation objectives. However, several concerns were raised regarding the scalability of the approach to high-dimensional action spaces, the reliance on dynamics models, the limited complexity of experimental tasks, missing or insufficient baselines such as behavior cloning, and aspects of clarity and presentation, particularly around partial observability and state construction.

In their response, the authors address the most central technical concern, dependence on a prior dynamics model, by adding new experiments demonstrating that MPAIL retains strong performance and out-of-distribution robustness when using fully online-learned dynamics. They also incorporate behavior cloning (BC) and behavior cloning from observation (BCO) baselines to better contextualize the learning-from-observation setting and clarify why access to expert actions materially simplifies imitation. Several reviewer questions are resolved through additional clarification of the problem setting, baselines, and method, and by moving key results such as computational benchmarks and OOD evaluations from the appendix into the main paper. Overall, the response strengthens the empirical case for the method and improves transparency varound assumptions and comparisons.

Remaining concerns primarily relate to scope and generality rather than correctness. The method’s scalability to higher-dimensional action spaces remains limited, as evidenced by weaker performance in more complex control settings, and it is unclear how well the approach would extend to domains such as manipulation without further algorithmic advances. The experimental evaluation, while carefully designed, is still focused on relatively simple tasks and a single real-world platform, leaving questions about broader applicability. Some reviewers may also still desire clearer attribution of performance gains across components, including planner, reward, value, and dynamics, and further refinement of presentation, particularly regarding partial observability handling. Overall, the response meaningfully addresses major reviewer concerns, but open questions about scalability and generalization to more complex domains remain.

**Reviewer Scores:**

The rebuttal most directly addresses the concerns of borderline reviewers who questioned the reliance on a prior dynamics model and the absence of simpler baselines, making 3 reviewers likely to update their scores upward. These reviewers already viewed the work as novel and practically relevant and explicitly raised issues that are resolved by the new learned-dynamics experiments and the added BC and BCO comparisons. In contrast, reviewers who focused primarily on presentation clarity or raised deeper concerns about scalability, computational cost, and high-dimensional control are less likely to change their evaluations.

---

### Decision · Program_Chairs · 2026-01-26

Accept (Poster)